# *Salmonella* multimutants enable efficient identification of SPI-2 effector protein function in gut inflammation and systemic colonization

Joshua P. M. Newson [1] ✉, Flavia Gürtler[1,3], Pietro Piffaretti[1], Nicolò Barbieri[1], Annina Meyer [1,4], Anna Sintsova[1], Manja Barthel[1], Yves Steiger [1], Sarah C. McHugh[1,5], Ursina Enz[1], Neal M. Alto [2], Shinichi Sunagawa [1] & Wolf-Dietrich Hardt [1] ✉

*Salmonella enterica* relies on translocation of effector proteins through the SPI-2 type III secretion system (T3SS) for pathogenesis. More than 30 effectors contribute to manipulation of host cells through diverse mechanisms, but interdependency or redundancy between effectors complicates the discovery of effector phenotypes using single mutant strains. Here, we engineer six mutant strains to be deficient in groups of SPI-2 effectors, as defined by their reported function. Using various animal models of infection, we show that three main phenotypes define the functional contribution of the SPI-2 T3SS to infection. Multimutant strains deficient for intracellular replication, for manipulation of host cell defences, or for expression of virulence plasmid effectors all show strong attenuation in vivo, while mutants representing approximately half of the known effector complement show phenotypes similar to the wild-type parent strain. By additionally removing the SPI-1 T3SS, we find groups of effectors that contribute to SPI-2 T3SS-driven enhancement of gut inflammation. Further, we provide an example of how iterative mutation can be used to find a minimal number of effector deletions required for attenuation, and thus establish that the SPI-2 effectors SopD2 and GtgE are crucial for promotion of gut inflammation and mucosal pathology.

A common virulence strategy of Gram-negative pathogens is the use of a type-three secretion system (T3SS) to translocate bacterial effector proteins into host cells[1,2]. Effectors provide a mechanism for the bacterial manipulation of host cells to produce outcomes that favour pathogenesis. *Salmonella enterica* serovars express two distinct T3SSs encoded on the genomic regions termed *Salmonella*-pathogenicity island-1 and −2 (SPI-1 and SPI-2)[3]. Collectively, effectors translocated by the SPI-1 T3SS mediate invasion into host cells and the induction of a strong inflammatory response in the gut lumen, which diminishes colonisation resistance and promotes expansion of luminal *Salmonella* populations, permitting robust transmission to new hosts[4–6]. In contrast, the SPI-2 T3SS is deployed exclusively by intracellular

[1]Institute of Microbiology, ETH Zurich, Zurich, Switzerland. [2]Department of Microbiology, University of Texas (UT) Southwestern Medical Center, Dallas, TX, USA. [3]Present address: Epidemiology, Biostatistics & Prevention Institute, University of Zurich, Zurich, Switzerland. [4]Present address: Institute of Food, Nutrition and Health, ETH Zurich, Zurich, Switzerland. [5]Present address: Department of Fundamental Microbiology, University of Lausanne, Lausanne, Switzerland. ✉e-mail: newsonj@ethz.ch; hardt@micro.biol.ethz.ch

*Salmonella*, which reside within the *Salmonella*-containing vacuole (SCV) in both epithelial cells and phagocytic immune cells[7,8]. SPI-2 effectors contribute to a range of phenotypes that promote intracellular replication and survival, which enables within-host migration to systemic niches and later reseeding of the gut[9,10]. Arguably, the SPI-1 and SPI-2 T3SS together represent the principal virulence factors that mediate the pathogenic lifestyle of *Salmonella*, and indeed a mutant deficient for both T3SS is greatly impaired at inducing gut inflammation, and is not able to efficiently invade into nor survive within host tissue[9,10]. These strong phenotypes should be attributable to the collective functions of translocated effector proteins, in addition to any effect exerted by the injection apparatuses themselves, and so it remains a central challenge to characterise the contribution of individual effectors. While most SPI-1 effectors have clear functions assigned, many SPI-2 effectors remain poorly characterised and it is not fully clear how individual effectors contribute to the virulence phenotypes mediated by the SPI-2 T3SS.

More than 30 SPI-2 T3SS effectors have been identified for the prototypical laboratory strain *S*. Typhimurium SL1344. Since the discovery and characterisation of the SPI-2 T3SS[7,8], decades of research has revealed how many of these effectors function to enable SPI-2 virulence, and these efforts are well reviewed elsewhere[11–13]. Briefly, SPI-2 effectors collectively contribute to a range of important intracellular activities, including the development and maintenance of the SCV, control of host cell trafficking, manipulation of cell signalling pathways that can lead to pro- or anti-inflammatory outcomes, and interference with the development of adaptive immunity[11,14]. Many SPI-2 effectors are enzymes that catalyse a diverse range of biochemical post-translational modifications to host proteins, while others act in a structural manner by binding to host enzymes to cause changes in substrate specificity[12]. The acquisition of a broad complement of effectors over evolutionary time likely contributes to the success of *Salmonella enterica* as a broad host-range pathogen, and similarly represents a highly tuneable bacterial strategy for keeping evolutionary pace with host cell defences that restrict bacterial proliferation. Thus, the study of SPI-2 effectors is critical to understand the mechanisms underpinning bacterial subversion of host cell processes, and should inform better strategies for control of this pathogen.

However, there are significant experimental challenges to the study of individual SPI-2 effectors. Logistically, the creation and characterisation of more than 30 single mutant strains is laborious, and while this strategy has been successful in screening for SPI-2 virulence phenotypes in vitro[15–17], there are significant experimental hurdles in more complicated experimental designs, for example in vivo experiments which require large numbers of animals or advanced imaging experiments which involve extensive sample preparation. Additionally, many effectors have been reported to have or are speculated to have redundant or interdependent functions. Previous studies exploring single mutant phenotypes have shown that single deletions for most effectors have no impact on intracellular replication or survival during in vitro experiments[15,18,19]. The creation of multimutant strains, in which more than one effector is deleted in a particular genetic background, has proven useful in identifying effectors that are necessary or sufficient for certain virulence phenotypes[15,20,21]. However, in many cases, the design of these multimutants precludes their use in systematically interrogating the function of all effectors in one experimental setup, as strong phenotypes arising from the deletion of certain effectors can mask the contributions of other deletions. Thus, there is a need for new tools that enable rapid and logistically simpler interrogation of SPI-2 effector functions, both to understand the activity of individual effectors and to explore how effectors cooperatively or redundantly contribute to broader virulence phenotypes.

Here, we describe the design and construction of six multimutant strains of *S*.Tm SL1344, each deficient for three to six different SPI-2 effectors, covering the known repertoire of SPI-2 effectors in this genetic background. We deployed these mutant strains in several murine models of *Salmonella* infection and found that effector cohorts required for intracellular replication and host-cell survival were critical for expansion in systemic niches, while approximately half of the SPI-2 effector repertoire remained dispensable for virulence. Further, these same effector cohorts were also required for migration from the gut to systemic niches during oral infection. We found that deletion of two effector cohorts could ablate the onset of SPI-2 T3SS-dependent gut inflammation, in the absence of SPI-1 T3SS effector translocation. Finally, we demonstrate a strategy for identifying the individual effectors that contribute to a cohort phenotype by use of simpler mutant strains, and thus describe an unreported role for the effectors SopD2 and GtgE in driving gut inflammation. Together, we show how complex multimutants can be deployed to interrogate virulence phenotypes, a strategy which should be broadly applicable in different experimental contexts.

## Results
### Construction and validation of SPI-2 effector multimutant strains
To surmount the experimental difficulties in studying how individual SPI-2 effectors contribute to various SPI-2 T3SS-mediated phenotypes, we designed and constructed a set of six multimutant strains in the *Salmonella* Typhimurium (*S*.Tm) SL1344 background. The design of these strains was guided by several criteria: effectors that contribute to similar phenotypes based on their reported functions should be deleted together; each effector should only be deleted once across all six strains; intergenic regions between closely located effectors should be preserved; and effectors should be removed as single deletions where possible, rather than deleting multiple closely-located effectors. The characterisation of SPI-2 effectors has been a priority since the identification and characterisation of the SPI-2 T3SS several decades ago, and thus the depth of literature available permits the loose grouping of effectors into functional groups (Fig. 1A). This informed the design of multimutant strains lacking effectors that could reasonably be deleted to produce a strain deficient for a particular function (e.g., a strain lacking several key effectors contributing to development and maintenance of the *Salmonella*-containing vacuole). Some effectors remain poorly characterised with no reported function or host targets, and these were similarly grouped as a multimutant strain. Finally, we used a single mutant deletion for *spvR* to represent a functional deletion of the virulence plasmid-encoded effectors *spvB, spvC*, and *spvD*, in line with previous work[15].

The workflow for generating multimutant strains (Fig. 1B) involved the initial construction of single-mutant strains representing all known SL1344 SPI-2 effectors, either by lambda red recombinase-mediated replacement of target genes with antibiotic resistance cassettes[22], or by leveraging an existing library of single mutant strains constructed in an *S*.Tm 14028S background[23] (see Table S1). These single mutant strains were then used to generate P22 lysates which contained randomly-packaged segments of the bacterial chromosome[24,25]. Using this library of P22 lysates, we sequentially performed P22 transduction to introduce these gene deletions into a clean SL1344 background, followed by Flp-FRT-mediated removal of resistance cassettes[26]. This process was repeated to sequentially delete up to six genes from an individual strain, and we performed this process in duplicate to generate two independent clone of each multimutant strain. Ultimately, we constructed six multimutant strains covering broad functional groups of *Salmonella* virulence, and assigned these strains designations based on the first six letters of the Greek alphabet (Fig. 1C). We performed whole-genome sequencing to validate the construction of these multimutant strains, and to determine the degree of unintended site variations on the chromosome of these strains (Fig. 1D). This analysis confirmed that all strains bear the correct deletion based on the intended design and are thus suitable for experimental use.

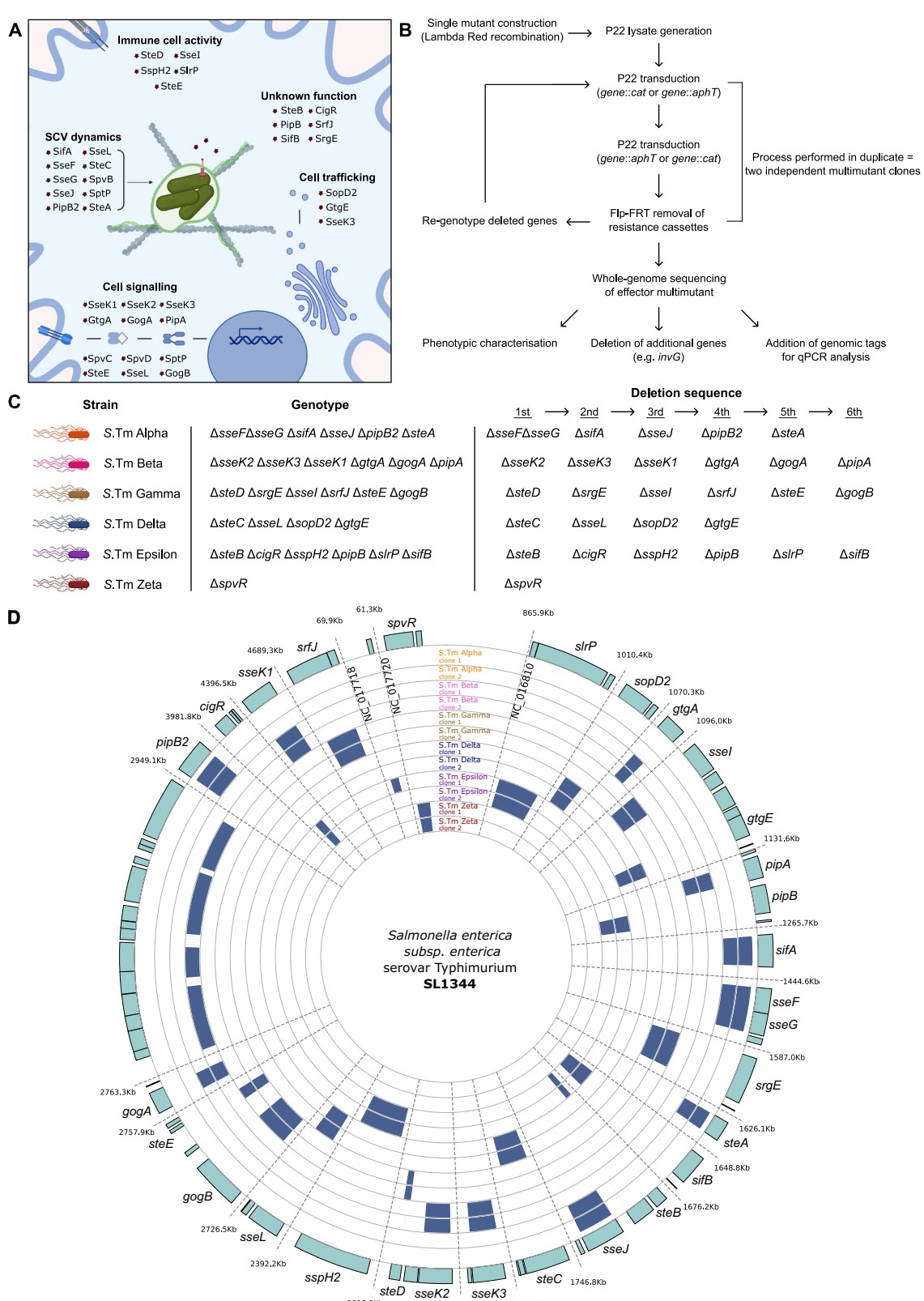

We detected a number of polymorphisms that likely arose due to successive genetic manipulation and passaging in laboratory conditions, which seemed to loosely correlate to the number of genes deleted in each strain (i.e., strains with six deletions tended to bear more polymorphisms than those with four) (Supplementary Data 1). Other polymorphisms are likely a consequence of transducing genes that were originally deleted in the S.Tm 14028 background and

transferred to the SL1344 genome (i.e., naturally occurring differences in the SL1344 and 14028 genomes) and these are annotated in Supplementary Data 1. We observed distinct polymorphisms arising between two independently constructed clones of each multimutant, suggesting these genetic changes arise randomly during the construction process. This also provides the opportunity to compare the fitness of each clone, to establish whether particular virulence

**Fig. 1 | Functional grouping of SPI-2 effector genes permits rationally designed multimutants. A** Graphic representation of reported functions for SPI-2 T3SS effector proteins. Effectors are loosely grouped into five different functional groups based on literature. Some effectors reportedly contribute to different functional groups and are represented twice here. Created in BioRender. Newson, J. (https://BioRender.com/leiik25). **B** Schematic representation of workflow to generate complex multimutants. Single mutants were created by lambda red recombination to replace genes of interest with cassettes encoding either kanamycin (::*aphT*) or chloramphenicol (::*cat*) resistance, followed by generation of P22 lysates containing phage that have packaged these deletions. Successive rounds of sequential P22 transduction and Flp-FRT removal of resistance cassettes resulted in mutants deficient for three to six effector genes, as required. Mutant strains were validated by whole genome sequencing to confirm deletion of target genes and to assess the degree of other changes to the genome. **C** Rational design of SPI-2

effector multimutant strains based on reported functions described in Fig. 1A. Six multimutant strains were constructed in duplicate and assigned designations based on the Greek alphabet (left) by deletion of effector genes in the sequence shown (right) to produce distinct multimutant genotypes (middle). Created in BioRender. Newson, J. (https://BioRender.com/j6182nu). **D** Bioinformatic analysis confirming the deletion of target genes from corresponding multimutant strains. Blue rectangles correspond to absence of reads mapping to chromosomal regions encoding these genes. High levels of single nucleotide polymorphisms can also produce an absence of reads mapping, as seen for *S*.Tm Beta clone 2 between *gogA* and *pipB2*, and for *S*.Tm Epsilon clone 1 between *srfJ* and *spvR*. Two independently-constructed multimutant strains were analysed. Gene position (top) corresponds to position on the chromosome of *S*.Typhimurium SL1344, while other regions of the chromosome are not shown.

phenotypes may be attributable to these polymorphisms. Finally, we observed no other small insertions or deletions on the chromosome, nor did we detect changes to the typical phage complement present in the reference genome. Thus, we report the correct construction of six SPI-2 effector multimutant strains that are suitable for use in various experimental contexts and should be useful in advancing the study of individual and collective SPI-2 effector-mediated phenotypes.

**SPI-2 effector cohorts contribute to systemic infection in vivo**
Next, we aimed to characterise the phenotypes of these multimutant strains in an infection context, and thus describe how different effector cohorts contribute to virulence. We infected C57BL/6 mice with $10^3$ *S*.Tm by intraperitoneal (i.p.) injection (Fig. 2A), which is a well-established murine model of systemic infection that is characterised by high levels of bacterial replication in systemic niches such as the spleen and liver[27,28]. To contrast attenuation caused by disruption of the SPI-2 T3SS to the virulence of *S*.Tm WT, we also infected mice with *S*.Tm Δ*ssaV* which is deficient for assembly of the SPI-2 T3SS, and with *S*.Tm Efl which is deficient for all known SPI-2 effectors but remains competent for SPI-2 T3SS assembly[15]. At day 4 post infection (Fig. 2B), we observed very high bacterial loads in the liver (left) and spleen (right) of mice infected with *S*.Tm WT, while mice infected with *S*.Tm Δ*ssaV* or *S*.Tm Efl had greatly reduced bacterial loads, consistent with the critical role played by the SPI-2 T3SS in mediating intracellular replication and survival[15,27]. Of the six multimutant strains, we observed a significant reduction in bacterial loads in mice infected with *S*.Tm Alpha, Delta, and Zeta, suggesting the effector groups deleted here play important roles in systemic infection. Surprisingly, three multimutant strains–*S*.Tm Beta, Gamma, and Epsilon–showed similar bacterial loads to the WT in both liver and spleen, suggesting these effectors are not required for virulence under these conditions. Further, we observed a similar trend in bacterial numbers in the mesenteric lymph node (Fig. S1A, left), suggesting these effectors play similar roles in colonisation of this site, while colonisation of the gut showed a less clear trend (Fig. S1A, middle) but nonetheless suggests the SPI-2 T3SS is important for delayed gut colonisation, consistent with previous findings[29]. These differences in recoverable CFU arise despite equivalent CFU in inocula delivered to mice by i.p. injection (Fig. S1A, right). Further, we also tested the second independently-constructed clones of each multimutant and observed broadly similar phenotypes (Fig. S1B). Finally, to determine if these strong phenotypes arise as a result of bacterial replication and survival, or if these differences are attributable to an initial failure to successfully colonise these sites, we repeated these experiments but euthanised mice at day 2 post infection. We observed a broadly similar trend at this earlier stage of infection (Fig. S1C) compared to later stages (Fig. 2B, and Fig. S1A), which suggests that while these functional cohorts do play distinct roles at early stages, the contribution of these effectors to virulence becomes significantly more pronounced as the systemic infection progresses. Thus, our set of multimutants can provide an efficient

means to survey which SPI-2 effectors contribute to a particular virulence phenotype.

A common strategy for studying virulence phenotypes is to perform competitive index experiments, in which a mixed inoculum comprising both the mutant strain and the wild-type strain is used for infection[30–32]. In animal models of infection, the testing of multiple strains in the same mouse reduces the number of animals required for the analysis, and provides internal controls for animal-specific differences in disease progression. To enable this approach using our multimutant strains, we introduced a unique fitness-neutral genetic tag into each strain[33], alongside the control strains *S*.Tm WT, *S*.Tm Δ*ssaV*, and *S*.Tm Efl. These tags can be quantified with a very high signal-to-noise ratio (greater than 1:100 000), either by quantitative RT-PCR or by a sequence counting utilising PCR amplification and next generation sequencing[34]. The resulting collection of 9 tagged strains was then used to infect C57BL/6 mice by i.p. infection (Fig. 2C) to assay the relative fitness of each strain within a single animal. We observed a similar trend as for single infection (Fig. 2B) in these mixed infection experiments (Fig. 2D, and Fig. S2 A, B), in which *S*.Tm WT, *S*.Tm Beta, *S*.Tm Gamma, and *S*.Tm Epsilon greatly outcompeted *S*.Tm Δ*ssaV*, *S*.Tm Efl, and the mutant strains *S*.Tm Alpha, *S*.Tm Delta, and *S*.Tm Zeta. We also tested the second clone of each multimutant bearing a different genetic tag and observed similar patterns of relative fitness (Figure. S2C). These data suggest the deficiencies of effector deletion strains cannot be compensated for by the presence of *S*.Tm WT or other mutants bearing WT-copies of effector genes within the same host animal.

**Virulence-dependent migration from the gut to systemic niches**
Oral infection represents the natural route of *Salmonella* infection in mice and other animals, and is characterised by invasion into epithelial tissue and induction of a strong inflammatory response in the gut lumen, followed by migration to systemic sites like the spleen and liver which serve as a niche for bacterial replication[3,35]. While the SPI-1 T3SS is the principal virulence factor that mediates gut infection, the SPI-2 T3SS also plays important roles in the colonisation of the lamina propria and a delayed but potent induction of gut inflammation[10,36]. Similarly, there is a strong requirement for SPI-2 T3SS activity in order for *S*.Tm to reach systemic niches beyond the gut[10,36]. To explore how particular SPI-2 effectors might contribute to these phenotypes, we infected mice (Fig. 3A) using the well-established streptomycin pre-treatment model of oral infection[10,37]. By day 4 post infection, *S*.Tm WT had colonised (Fig. 3B) both the liver (left) and spleen (right) and replicated to high numbers, while both *S*.Tm Δ*ssaV* and *S*.Tm Efl were recovered either at very low numbers or not at all, indicating a strong reliance on SPI-2 T3SS virulence for invasive infection of systemic niches. We observed lower population sizes in mice infected with either *S*.Tm Alpha or *S*.Tm Zeta, while *S*.Tm Delta showed an especially pronounced reduction in bacterial load, similar to that observed for *S*.Tm Δ*ssaV* and *S*.Tm Efl. In contrast, we observed less pronounced

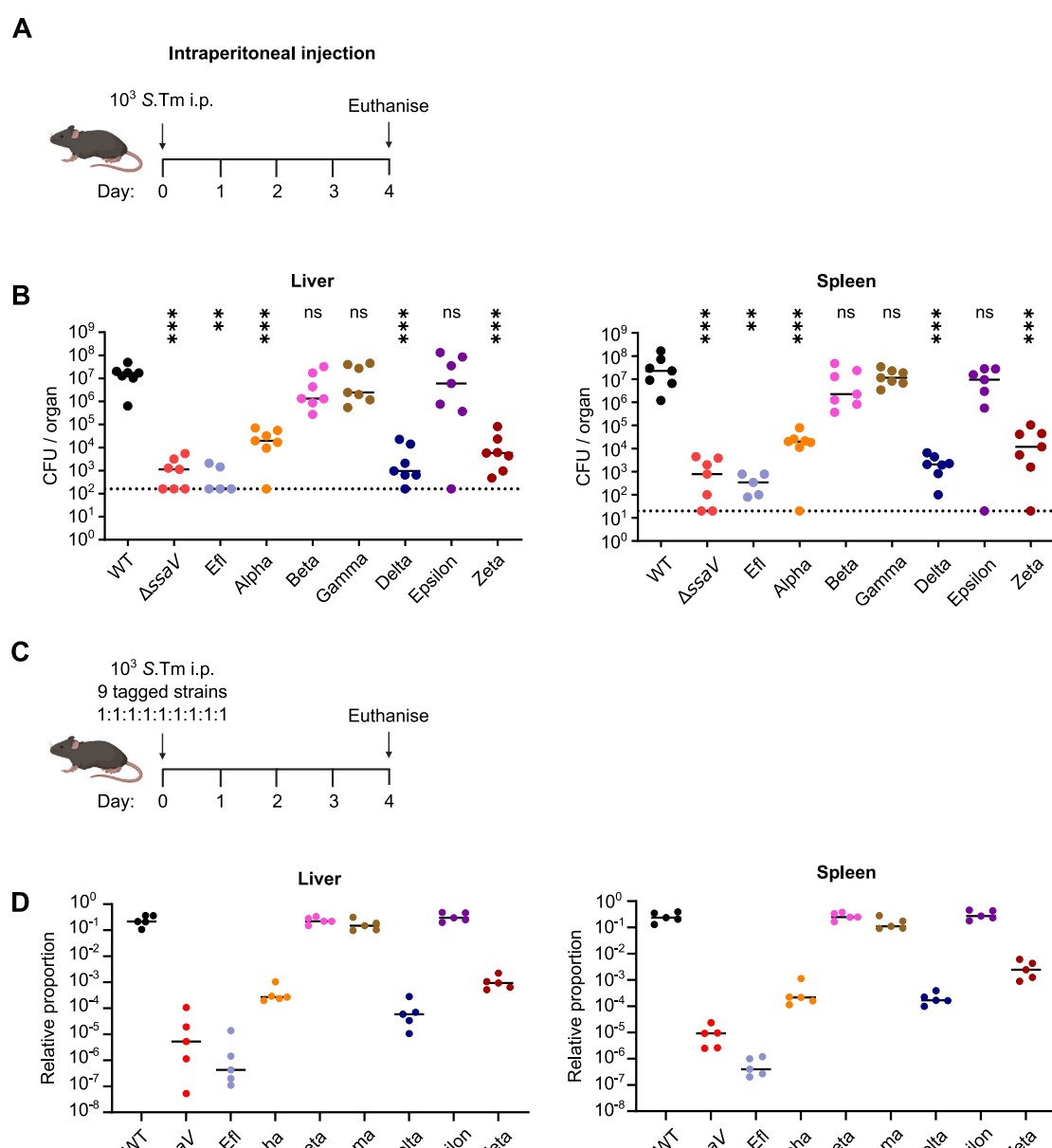

**Fig. 2 | Systemic infection is compromised by deletion of SPI-2 effector cohorts.** **A** Experimental scheme to study virulence of *S*.Tm mutants during systemic infection in vivo. Mice were infected with $10^3$ *S*.Tm by intraperitoneal injection. Mice were euthanised at day 4 post infection, and bacterial loads in the spleen and liver were quantified by CFU plating to selective media. Created in BioRender. Newson, J. (https://BioRender.com/ngy8g3l). **B** *S*.Tm recovered from the liver (left) and spleen (right), (*n* = 5–7 mice per group). Horizontal bars denote median. Dotted lines denote limit of detection. Statistical differences between WT and indicated groups determined by two-tailed Mann Whitney-U test, (*p* ≥ 0.05 not significant

(ns), *p* < 0.01 (**), *p* < 0.001 (***). **C** Experimental scheme to study relative fitness of *S*.Tm mutants in vivo by competitive infection. Mice were infected with a mixed inoculum comprising equal volumes of 9 different *S*.Tm strains each bearing unique chromosomal tags. Mice were euthanised at day 4 post infection. Created in BioRender. Newson, J. (https://BioRender.com/9d0jurp). **D** Relative proportion of each genetic tag determined by RT-qPCR. Data is presented as the proportion of a given tag relative to the other tags within one animal. Coloured circles represent tagged strain recovered from the liver (left) and spleen (right) of infected mice (*n* = 5 mice). Horizontal bars denote median.

differences in the mesenteric lymph node (Fig. 3C), where only *S*.Tm Δ*ssaV* and *S*.Tm Delta showed modestly reduced CFU. Similarly, we recovered equivalent numbers in the faeces of mice infected with each strain (Fig. 3D), indicating no significant contribution to gut luminal populations as expected for this model[10]. These phenotypes arise despite equivalent CFU in the inocula (Fig. S3A), and we observed no invasion defect for any multimutant strain during control experiments performed in vitro (Fig. S3B). Finally, we measured enteropathy in the caecum tissue and observed a broadly similar degree of pathology in

all mice (Fig. 3E, F). While this may suggest a limited contribution of SPI-2 effectors to gut pathology, it seems more likely that the strong pathology induced by the SPI-1 T3SS[10,37] masks more subtle contributions by SPI-2 effectors. Overall, these data suggest that while there are minimal differences in gut colonisation and tissue pathology between multimutant strains, the subsequent migration to systemic niches followed by intracellular replication and survival is strongly dependent on particular cohorts of SPI-2 effectors, while others remain surprisingly dispensable.

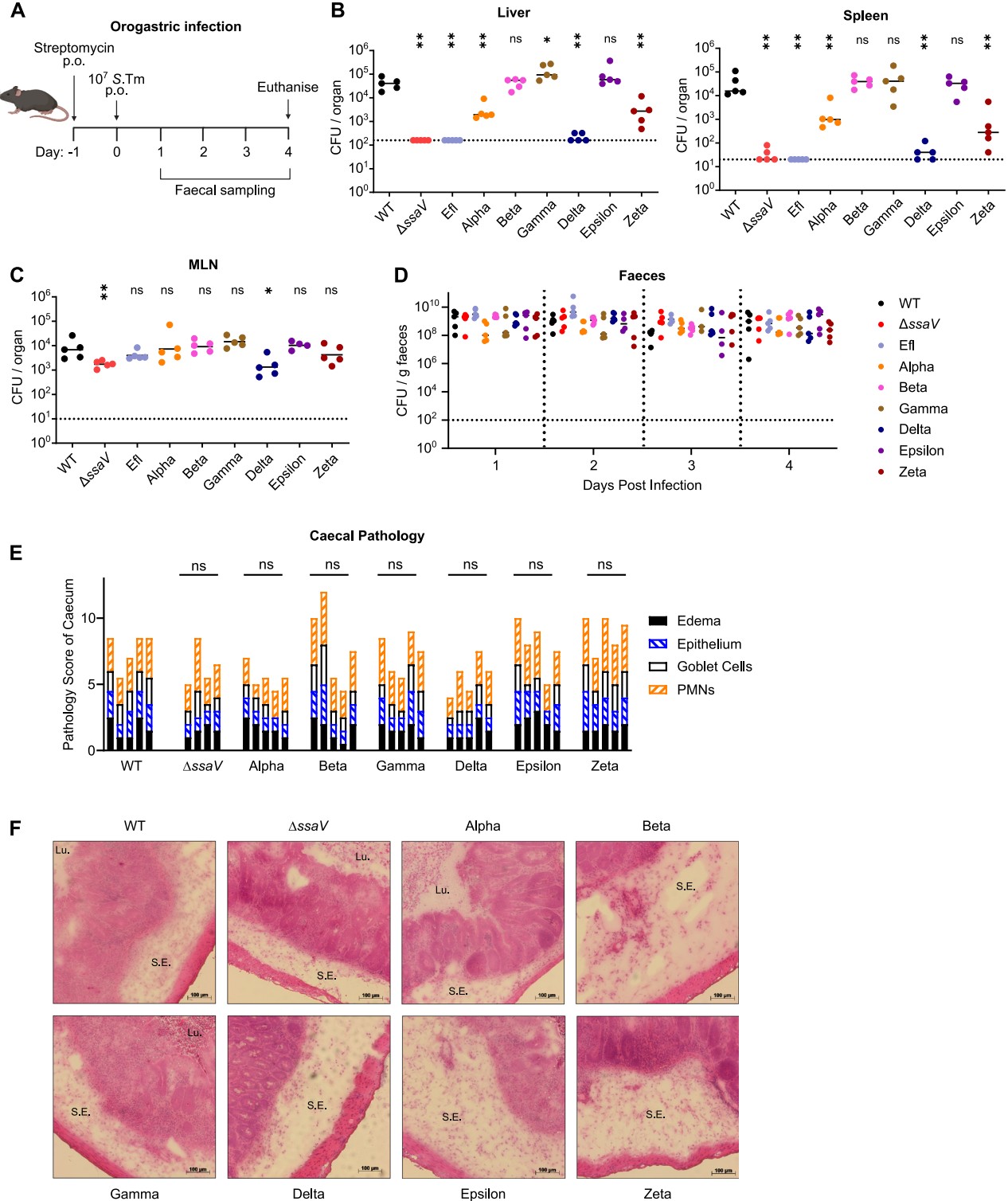

**Fig. 3 | SPI-2 effectors support bacterial migration from the gut to systemic niches. A** Experimental scheme to study virulence of *S.*Tm mutants during oral infection in vivo. Mice were pre-treated with streptomycin by oral gavage, then received an infectious dose of 5 × 10⁷ *S.*Tm by oral gavage. Faeces were collected at indicated time points and mice were euthanised at day 4 post infection. Created in BioRender. Newson, J. (https://BioRender.com/spwhgw9). **B, C** Bacterial loads recovered from (**B**) the liver (left), spleen (right) and (**C**) mesenteric lymph node at day 4 post infection (*n* = 5 mice per group). Dotted lines denote limit of detection. Horizontal bars denote median. **D** Bacterial populations in the gut determined by CFU plating of homogenised faecal samples to selective media. Dotted line at 10²

CFU / g faeces denotes conservative limit of detection (*n* = 5 mice per group). Horizontal bars denote median. **E, F** Caecal histology at day 4 post infection. **E** combined pathology score based on scoring criteria quantifying submucosal oedema, epithelial barrier integrity, goblet cell number, and infiltration of poly-morphonuclear granulocytes. **F** representative micrographs of cecum samples stained with hematoxylin and eosin. Lu. lumen, S.E. submucosal oedema. Scale bars indicate length of 100 μm. Statistical differences between WT and indicated groups determined by two-tailed Mann Whitney-U test, (*p* ≥ 0.05 not significant (ns), *p* < 0.05 (\*), *p* < 0.01 (\*\*).

## Induction of SPI-2 T3SS-driven gut inflammation requires cohorts of effectors

Using our SPI-2 multimutant strains, we observed relatively little difference in the induction of enteropathy in the caecal tissue (Fig. 3E-F), which is a hallmark of oral infection in mice and driven largely by the SPI-1 T3SS[10,37,38]. We speculated that more subtle contributions of SPI-2 effectors to gut infection, particularly the delayed onset of inflammation, might be masked by the activity of SPI-1 effectors. To explore this, we created a set of SPI-2 multimutant strains that is additionally deficient for SPI-1 T3SS effector translocation by deleting *invG*, encoding a key structural component of the SPI-1 T3SS[39]. Thus, these strains are deficient for SPI-1 effector translocation and additionally lack genes for distinct cohorts of SPI-2 effectors, as in Fig. 1C. These tools allow for the elucidation of subtle phenotypes that are otherwise undetectable against the severe gut pathology induced by SPI-1 effectors. We performed oral infection in streptomycin pre-treated mice as previously (Fig. 3A) and observed that both WT and *S*.Tm Δ*invG* are recovered at similar numbers in the liver, spleen, and mesenteric lymph node by day 4 post infection (Fig. 4A), consistent with previous work[10,37]. Concordantly, there was little difference in CFU recovered for *invG*-deficient multimutants at these sites (Fig. 4A) compared to the respective *invG*-competent strains (Fig. 3B, C). This data agrees with reports that SPI-2 remains the primary virulence factor mediating colonisation of systemic sites during orogastric infection[10,36,40]. *S*.Tm loads in the faeces were high for all strains at day 1 post infection (Fig. 4B), consistent with equivalent CFU in the inocula (Figure. S3C) and previous work using streptomycin pretreatment of mice[10,37]. By day 4 post infection, we observed a trend towards reduced faecal loads for *S*.Tm Δ*invG*Δ*ssaV* (deficient for both SPI-1 and SPI-2 T3SS assembly), consistent with previous reports[41]. Interestingly, a similar trend was observed for all *invG*-deficient multimutants (Fig. 4B), perhaps suggesting previously unidentified roles for diverse SPI-2 T3SS effectors in the prolongation of *S*.Tm gut colonisation.

While induction of gut inflammation mediated by the SPI-1 T3SS is a well-established hallmark of infection in the streptomycin pretreatment model[37], nevertheless SPI-1 mutants can still cause delayed but significant inflammation in a SPI-2 T3SS-dependent manner[36]. While the mechanisms of SPI-2 driven inflammation remain enigmatic, this delayed inflammation has been linked to prolonged *S*.Tm gut colonisation[10,41–43]. Here, we also observed a gradual but strong increase in gut inflammation in the *S*.Tm Δ*invG* strain (thus caused by SPI-2 effectors), while *S*.Tm Δ*invG*Δ*ssaV* strain fails to induce gut inflammation, as expected (Fig. 4C). For most *invG*-deficient multimutants, we observed a similar trend in which initially uninflamed conditions in the gut gave rise to potent inflammation by day 4 post infection, based on lipocalin-2 ELISA. However, we noted a previously unappreciated trend in which no multimutant produced inflammation to the degree of the SPI-1 or SPI-2 competent strains, which may suggest that diverse perturbations of the SPI-2 effector complement can disrupt the induction of gut inflammation. Regardless, we observed particularly strong phenotypes for *S*.Tm Alpha and *S*.Tm Delta, which showed inflammation profiles similar to that of *S*.Tm Δ*invG*Δ*ssaV* and greatly diminished relative to *S*.Tm WT (Fig. 4C). To further explore the contributions to gut pathology, we repeated our examination of caecal pathology in these mice (Fig. 4D, E). Here, we observed that both *S*.Tm Δ*invG* and *S*.Tm Δ*ssaV* could produce enteropathy that contributes to the strong level of disease seen in *S*.Tm WT-infected mice, while mice infected with *S*.Tm Δ*invG*Δ*ssaV* retained relatively healthy gut tissue. This healthy state was phenocopied by both *S*.Tm Alpha Δ*invG* and *S*.Tm Delta Δ*invG*, suggesting that in the absence of the SPI-1 T3SS these effector cohorts contribute to potent SPI-2-dependent gut inflammation and pathology. All other *invG*-deficient multimutants produced enteropathy approaching that of the control strains, indicating a dispensability for the induction of gut pathology. These data provide insights into how cohorts of SPI-2 effectors can contribute to delayed but significant inflammatory phenotypes in the intestinal mucosa.

## Iterative deletion of effector genes reveals individual SPI-2 effectors required for gut inflammation

As described in Fig. 1C, the multimutant strains were constructed via sequential deletion of individual effector genes. Thus, the creation of a six-fold mutant required the preceding construction of the parent five-fold mutant, and before that a four-fold mutant, *et cetera*. Each stepwise mutant was preserved in cryostorage, and thus it is possible to use these simpler mutants to determine which particular genes may contribute to the phenotype observed for the complete multimutant. To provide an example of this strategy, we chose to focus on the *S*.Tm Delta strain, which we showed was deficient for colonisation of systemic niches during intraperitoneal infection (Fig. 2), and showed pronounced attenuation for migration from the gut to these systemic niches in oral infection (Fig. 3), and was ultimately shown to contribute to SPI-2-dependent gut inflammation and enteropathy (Fig. 4). The *S*.Tm Delta mutant comprises deletions in four effector genes: *steC*, *sseL, sopD2*, and *gtgE*. To determine which effectors contribute to the strong phenotype observed for this multimutant, we used the Δ*steC*Δ*sseL* double mutant created during the stepwise construction of *S*.Tm Delta, and separately constructed a Δ*sopD2*Δ*gtgE* double mutant. We infected mice by oral gavage as previously (Fig. 3A), and observed that *S*.Tm Δ*steC*Δ*sseL* phenocopied *S*.Tm WT, while the Δ*sopD2*Δ*gtgE* double mutant was recovered in similar numbers to *S*.Tm Δ*ssaV* (Fig. 5A, and Fig. S4A), and thus these two effectors alone mediate the *S*.Tm Delta phenotypes. We next used single mutants to determine the relative contribution of each individual effector, and found only partial reductions relative to the WT, demonstrating that both effectors must be deleted together to produce this phenotype, likely due to the functional overlap between these effectors. Finally, we complemented the Δ*sopD2*Δ*gtgE* double mutant by sequential chromosomal restoration of WT copies of these genes, and found that the double-complemented Δ*sopD2*Δ*gtgE* mutant was restored to approximately WT levels in this infection model (Fig. 5A, and Fig. S4A).

Previous work has described how SopD2 and GtgE are critical for systemic proliferation following intraperitoneal injection of mice[44], but the contributions of these effectors to gut pathology remains unexplored. Here, we established that gut inflammation during oral infection could be ablated by deletion of both the SPI-1 T3SS and certain cohorts of SPI-2 effectors, including those deficient in *S*.Tm Delta (Fig. 4D, E). Given that deletion of *sopD2* and *gtgE* was sufficient to phenocopy *S*.Tm Delta in terms of systemic colonisation (Fig. 5A), we hypothesised that deletion of these two effectors and the SPI-1 T3SS would similarly be sufficient to reduce gut inflammation to levels seen for the avirulent *S*.Tm Δ*invG*Δ*ssaV*. To explore this, we deleted *invG* from the double mutant *S*.Tm Δ*sopD2*Δ*gtgE* and orally infected mice as previously (Fig. 3A). Indeed, while a triple mutant *S*.Tm Δ*invG*Δ*sopD2*Δ*gtgE* failed to successfully colonise the liver, spleen, and mesenteric lymph nodes in a manner similar to the double mutant *S*.Tm Δ*sopD2*Δ*gtgE* (Fig. 5B), there was a marked decrease in gut inflammation whereby the triple mutant failed to induce both early and late stage inflammation, similar to levels seen for the avirulent Δ*invG*Δ*ssaV* strain (Fig. 5C, and Fig. S4B).

Intracellular reservoirs of *S*.Tm residing in the caecal tissue can contribute to sustained gut pathology[10,45]. We performed a gentamicin protection assay on infected mucosal tissue[46] to determine the contribution of SopD2 and GtgE to survival of intracellular *S*.Tm within caecal tissue. Here, we recovered fewer *S*.Tm Δ*invG*Δ*sopD2*Δ*gtgE* and *S*.Tm Δ*invG*Δ*ssaV* relative to *S*.Tm WT (Fig. 5D), perhaps suggesting local replication and survival in caecal tissue is important for sustained gut inflammation. Finally, we discovered a corresponding ablation of enteropathy in the cecum of mice infected with either *S*.Tm Δ*invG*Δ*sopD2*Δ*gtgE* or Δ*invG*Δ*ssaV*, confirming that deletion of these

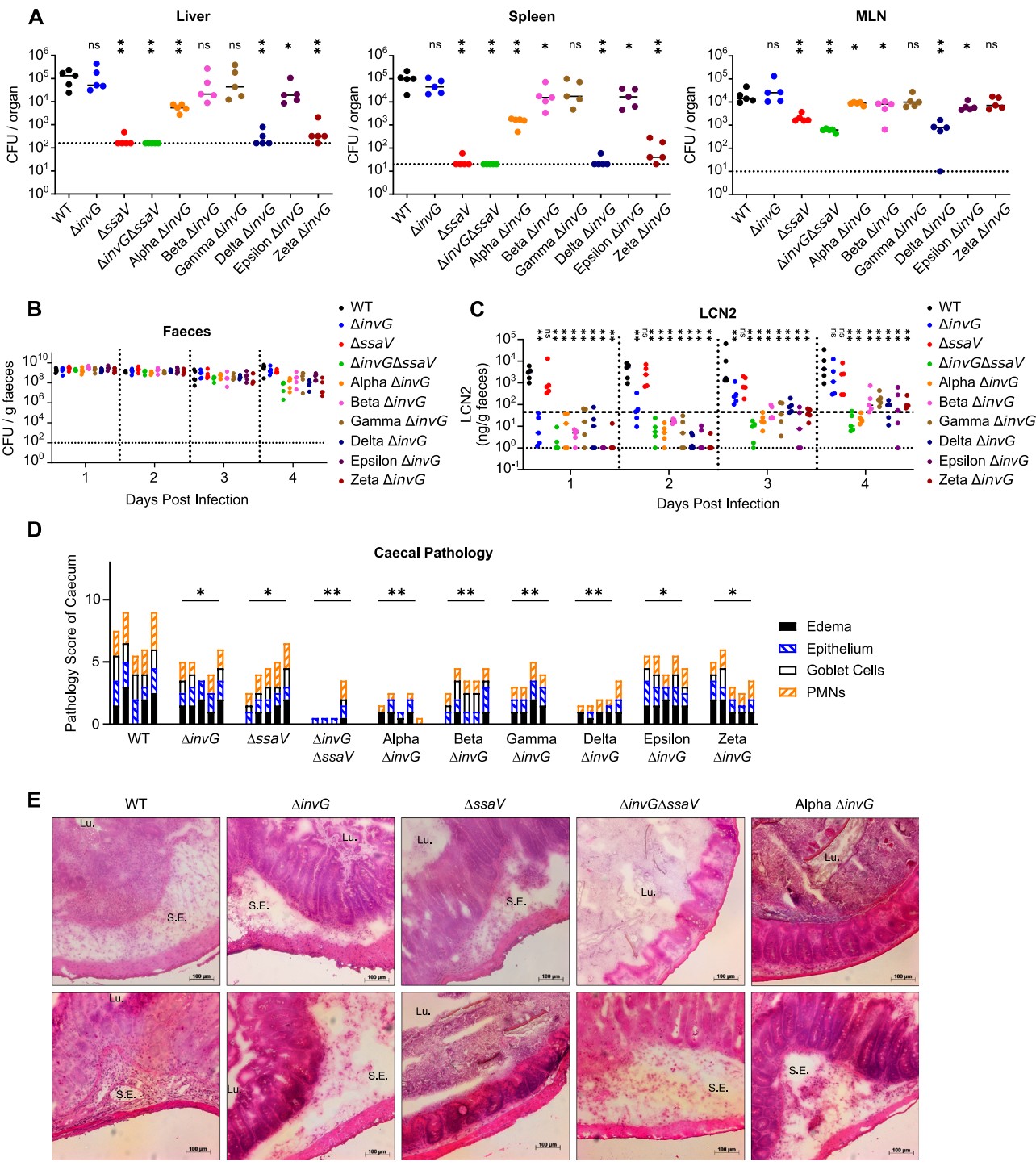

**Fig. 4 | Effector cohorts contribute to SPI-2 T3SS-dependent inflammation.**
**A** Bacterial loads recovered from the liver (left), spleen (middle), and mesenteric lymph node (right) at day 4 post infection ($n = 5$ mice per group). Mice were infected as in Fig. 3A. Dotted lines denote limit of detection. Horizontal bars denote median. **B** Bacterial populations in the gut determined by CFU plating of homogenised faecal samples to selective media. Dotted line at $10^2$ CFU / g faeces denotes conservative limit of detection ($n = 5$ mice per group). Horizontal bars denote median. **C** Levels of gut inflammation at indicated days post infection determined by ELISA quantification of lipocalin-2 (LCN2). Horizonal dotted line (upper) represents typical threshold of moderately inflamed gut, while dotted line (lower) represents limit of detection. Horizontal bars denote median. **D, E** Caecal histology at day 4 post infection. **D** combined pathology score based on scoring criteria quantifying submucosal oedema, epithelial barrier integrity, goblet cell number, and infiltration of polymorphonuclear granulocytes. **E** representative micrographs of cecum samples stained with hematoxylin and eosin. Lu. lumen, S.E. submucosal oedema. Scale bars indicate length of 100 μm. Statistical differences between WT and indicated groups determined by two-tailed Mann Whitney-U test, ($p \geq 0.05$ not significant (ns), $p < 0.05$ (*), $p < 0.01$ (**).

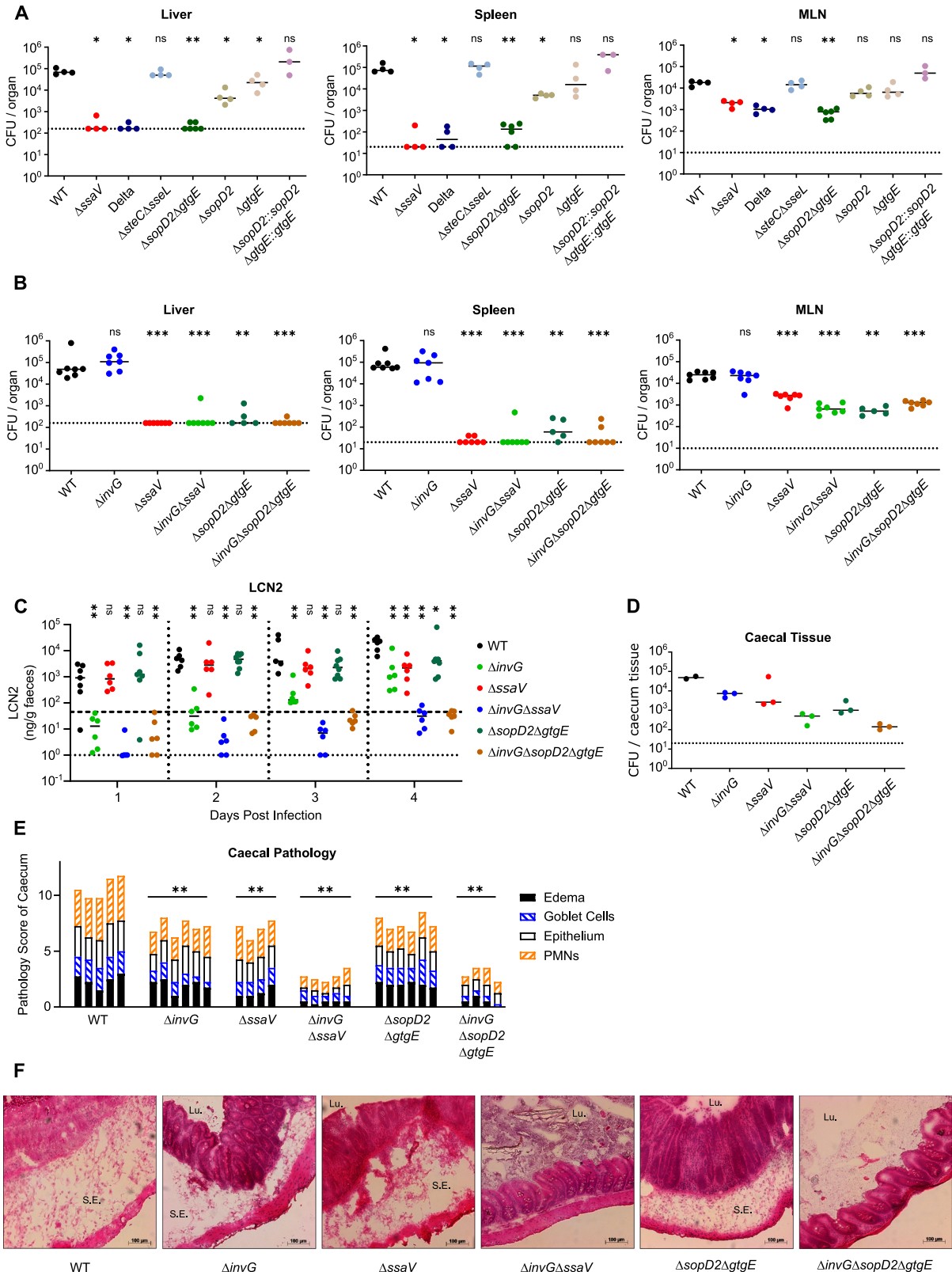

two SPI-2 effectors is sufficient for ablation of SPI-2 T3SS-dependent gut inflammation (Fig. 5E, F). Collectively, these data provide new insights into how specific SPI-2 effectors contribute to inflammatory outcomes in the infected gut (Fig. 6), and provide a proof of concept for how strong phenotypes observed using *S*.Tm effector multimutant strains can be rapidly narrowed to candidate effectors responsible for this activity.

## Discussion

The intracellular lifestyle of pathogenic *Salmonella enterica* is driven by the activity of SPI-2 T3SS effectors, but efforts to characterise the function of individual effectors have proven complicated, either by instances of interdependency or redundancy between effectors, or by the logistical difficulties in characterising more than 30 proteins across different experimental contexts. Here, we designed SPI-2 effector

**Fig. 5 | SPI-2 effectors SopD2 and GtgE contribute to SPI-2 T3SS-dependent inflammation. A**, **B** Bacterial loads in the liver (left), spleen (middle), and mesenteric lymph node (right) at day 4 post infection ($n$ = 3-6 mice per group). Mice were infected as in Fig. 3A. Dotted lines denote limit of detection. Horizontal bars denote median. **C** Levels of gut inflammation at indicated days post infection determined by ELISA quantification of lipocalin-2 (LCN2). Horizonal dotted line (upper) represents typical threshold of moderately inflamed gut, while dotted line (lower) represents limit of detection. Horizontal bars denote median ($n$ = 6-8 mice per group). **D** Intracellular populations of bacteria recovered from caecal tissue at day 4 post infection by gentamicin protection assay ($n$ = 2-3 mice per group). Dotted lines denote limit of detection. Horizontal bars denote median. **E**, **F** Caecal histology at day 4 post infection. **E** combined pathology score based on scoring criteria quantifying submucosal oedema, epithelial barrier integrity, goblet cell number, and infiltration of polymorphonuclear granulocytes. **F** representative micrographs of cecum samples stained with hematoxylin and eosin. *Lu.* lumen, *S.E.* submucosal oedema. Scale bars indicate length of 100 μm. Statistical differences between WT and indicated groups determined by two-tailed Mann Whitney-U test, ($p \geq 0.05$ not significant (ns), $p < 0.05$ (*), $p < 0.01$ (**), $p < 0.001$ (***).

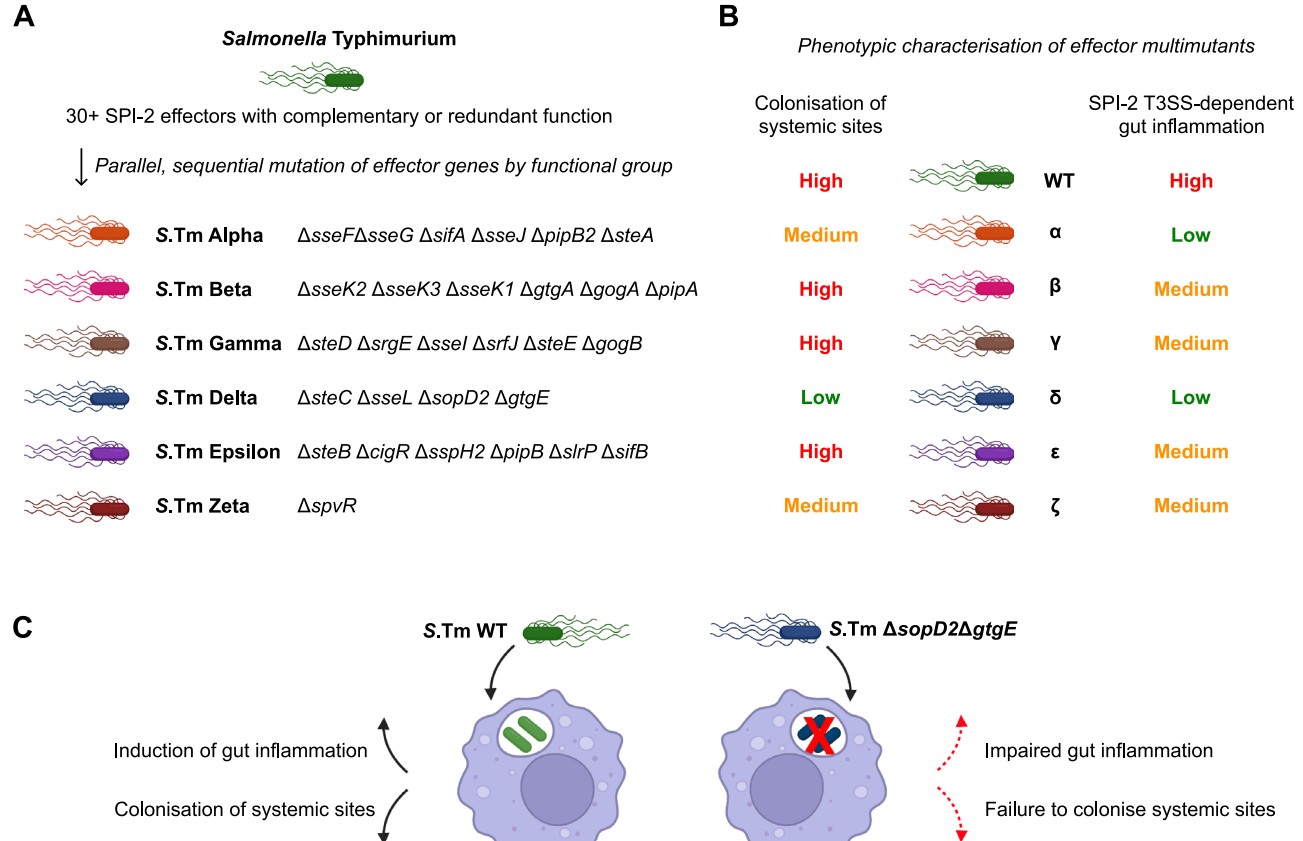

**Fig. 6 | Graphical summary of effector cohort contributions to inflammation and systemic colonisation. A** Genotypes of six SPI-2 T3SS effector multimutants generated by successive rounds of P22 transduction from single mutant backgrounds. Collectively, these strains cover all described SPI-2 effectors in the SL1344 background. **B** Summary of phenotypic data described in Figs. 2–4. *S*.Tm WT causes high levels of gut inflammation and migrates from the gut to colonise systemic tissue, in a manner dependent on the SPI-2 T3SS. Several multimutants show similar levels of virulence to WT. *S*.Tm Alpha, Delta, and Zeta showed reduced colonisation of systemic niches, while SPI-2 T3SS-dependent inflammation was particularly impaired during infection with *S*.Tm Alpha and Delta. **C**) *S*.Tm WT relies on intracellular niches to induce inflammation and achieve colonisation of systemic tissue. A mutant deficient for SopD2 and GtgE fails to maintain the intracellular niche, leading to reduced levels of gut inflammation and impaired colonisation of systemic sites. Created in BioRender. Newson, J. (https://BioRender.com/ojbf1us).

multimutants to explore which effector cohorts are critical for pathogenesis in a logistically easier manner, and showed how such tools can be used to find a minimal set of effectors responsible for key phenotypes, using SPI-2-dependent promotion of gut inflammation as an example.

In designing and constructing SPI-2 effector multimutants, we loosely grouped effectors that reportedly contribute to similar functions. For example, the six effectors deleted in *S*.Tm Alpha (*sseF, sseG, sifA, sseJ, pipB2, steA*) collectively contribute to development and maintenance of the SCV, while *S*.Tm Beta is deficient for six effectors (*sseK1, sseK2, sseK3, gtgA, gogA, pipA*) that antagonise different aspects of host cell signalling pathways. The rational grouping of deletions is dependent on the reported function of each effector (Fig. 1A) (reviewed in refs. 12,13), and many effectors have functions that are

unknown or disputed. Thus, it is possible that unreported functions or inter-effector relationships may contribute to phenotypes that are masked by the current design. Similarly, undiscovered effectors likely exist and the activity of those effectors may contribute to shared phenotypes. While we observed strong phenotypes for several multimutants (*S*.Tm Alpha, Delta, and Zeta), other mutants (*S*.Tm Beta, Gamma, and Epsilon) representing approximately half of the known SL1344 effector repertoire phenocopied *S*.Tm WT in different infection models. It is possible that these effectors do have phenotypes in other infection models (e.g., in a different host species, or during chronic infection), or that the impact is too subtle to measure via the methods used here. Alternatively, more pronounced phenotypes may emerge when particular effector deletions are grouped together in different combinations to those constructed here. Ultimately, further careful

characterisation of these mutants in different infectious contexts and using different methodology will be useful to fully characterise the role of these effectors. An alternative explanation for the lack of strong phenotypes for several mutants tested here is the context-dependency of deletion mutants, described elsewhere as the effector network hypothesis[47–49]. Effectors of the gut pathogen *Citrobacter rodentium* reportedly form robust networks that can tolerate the loss of a number of effectors, but deletion of an increasingly large number of effectors ultimately causes a collapse of virulence phenotypes back to an avirulent level. Importantly, this same study describes context-dependent essentiality of effectors, in which deletion of a single effector may or may not produce a strong phenotype depending on the availability of other effectors[47]. Certainly, this possibility may also exist for the effector cohort of *S*.Typhimurium, and this may complicate the comparison of studies such as ours with previous and future work. More broadly, the study of how complex genetic interactions produce virulence phenotypes remains an important challenge for modern bacterial genetics. We anticipate that new advances in systems biology and network dynamics will be well-supported by experimental data describing phenotypes for multimutant strains, as shown for *Salmonella* spp. (here, and in ref. 15) and for other pathogens with extensive effector repertoires[47,50,51].

Indeed, other studies have employed the strategy of sequentially deleting multiple genes encoding *Salmonella* effector proteins, though the design and rationale for these efforts varies. Chen et al.[15] iteratively deleted the majority of known SPI-2 effectors in a single genetic background, producing an 'effectorless' *S*.Tm SL1344 derivative that is otherwise competent for SPI-2 T3SS assembly and function. Restoring selected effectors to this effectorless strain by complementation allowed for the identification of a 'minimal network' of effectors that was sufficient for virulence during oral infection (*sifA*, *sseFG*, *steA*, *sopD2*, and *spvBCD*)[15]. Elsewhere, separate studies have focused on deleting core sets of effectors to produce a strain that is attenuated to Δ*ssaV*-levels of virulence. Strong phenotypes have thus been reported for a seven-fold deletion strain (*S*.Tm Δ*sseF* Δ*sseG* Δ*sifA* Δ*sopD2* Δ*sseJ* Δ*steA* Δ*pipB2*)[20], and separately for a five-fold deletion (*S*.Tm Δ*sifA* Δ*spvB* Δ*sseF* Δ*sseJ* Δ*steA*, created in a SPI-1 T3SS-deficient background)[21]. These efforts represent important steps in understanding how individual effectors contribute to strong collective phenotypes, but they are less useful in contexts where screening for individual effectors is important. The advantage of our strategy described here is that the activity of all effectors can be explored in a single experimental context, and responsible effectors can subsequently be identified by use of simpler mutants. An important caveat to our approach is the use of particular single mutants originally constructed in an *S*.Tm 14028S background prior to transduction into *S*.Tm SL1344, and while this is considered acceptable practice in the field[52–57], this can result in the concomitant transfer of genetic elements from 14028S that are distinct from those found in SL1344. Ultimately, we anticipate that these multimutant strains could be used to screen for SPI-2 T3SS effector functions in other experimental contexts and infection models, for example to study chronic carriage in genetically resistant mice, intracellular replication in vitro in various host cell types, virulence phenotypes in various genetically-modified mouse backgrounds, or performance in reporter assays that measure cell signalling outcomes.

Our work highlights the strong contribution of SopD2 and GtgE to virulence in both oral and systemic infection. The molecular target of both of these effectors is the host GTPase Rab32[44,58], which restricts intracellular bacteria via its nucleotide exchange factor BLOC-3[59]. The acquisition of SopD2 and GtgE by *S*.Typhimurium permits the complementary antagonism of Rab32, in which SopD2 functions as a GAP mimic to limit Rab32 GTPase activity, while GtgE directly proteolytically cleaves Rab32[44,58]. In the absence of these effectors, Rab32 and the co-factor BLOC-3 facilitate the delivery of itaconate to the SCV[60,61],

which restricts intravacuolar *S*.Tm by metabolic disruption of the glyoxylate shunt and thereby reduces bacterial replication[62]. Thus, effector-mediated disruption of the Rab32-BLOC-3-itaconate axis represents an important strategy for the success of intracellular *S*.Tm populations. While it had been established that SopD2 and GtgE were important for promoting intracellular survival in systemic niches[44], the contributions of these effectors to gut pathology and inflammation remained unknown.

Here, we show that *S*.Tm Delta and *S*.Tm Δ*sopD2*Δ*gtgE* cannot reach systemic niches following oral infection (Fig. 3), and this mutant also show severe attenuation in systemic sites during intraperitoneal infection (Fig. 2), similar to *S*.Tm Δ*ssaV*[10,36]. Given that itaconate-mediated disruption of intracellular *S*.Tm restricts actively replicating bacteria[62], this may suggest that bacterial replication is an important activity for successful migration from the gut to systemic niches, or that cell-intrinsic host defences impose particularly stringent control of intracellular bacteria during these migration events or during subsequent bacterial growth in systemic niches. Importantly, our current work extends previous knowledge by discovering a previously unknown function of SopD2 and GtgE in eliciting mucosal inflammation in a SPI-1 T3SS-deficient background (Figs. 4C–E, and 5C–F). This finding will enable future work at molecular and cellular scales to explore why the *S*.Tm Δ*invG*Δ*sopD2*Δ*gtgE* mutant is drastically impaired at inducing gut inflammation. It may be that a reduction of bacterial numbers in the gut tissue causes a corresponding impairment of gut inflammation, and indeed we observed a modest reduction in CFU in the mesenteric lymph node (Fig. 5B) and faeces (Figure. S4B), and similarly a slight reduction in the intracellular population within caecal tissue (Fig. 5D). However, we cannot exclude the possibility that these effectors trigger a pro-inflammatory signalling response that promotes inflammation and enteropathy in the gut. Future work should focus on the molecular mechanisms that underpin how SopD2 and GtgE promote *S*.Tm growth and survival in the gut, on how this affects inflammation and mucosal pathology, and on how host cell defences (e.g., itaconate) prevent pathogen migration to systemic sites.

While we observed the strongest deficiency for *S*.Tm Delta, we also found that both *S*.Tm Alpha and *S*.Tm Zeta were significantly reduced in the liver and spleen during both oral (Fig. 3) and systemic infection models (Fig. 2). We observed that *S*.Tm Alpha colonised these sites at approximately *S*.Tm Δ*ssaV* levels by day 2 post infection (Figure. S1C), then increased modestly in the following days (Fig. 2B), suggesting this mutant can still replicate intracellularly to some extent, despite lacking the principal effectors mediating SCV maturation and expansion. Alternatively, it may be possible that this increase is attributable to extra-vacuolar or extracellular replication, or to cell-to-cell spread via efferocytosis or simply bacterial egress and reinvasion[63–65]. Future work, especially using in vitro models of infection, will be useful to determine the replicative defect of this strain. We observed that *S*.Tm Alpha Δ*invG* was greatly attenuated at inducing mucosal inflammation by day 4 post infection (Fig. 4C), similar to *S*.Tm Delta Δ*invG*, and this is consistent with previous work linking some of these effectors to gut inflammation[21]. Future work is needed to understand which minimal subset of effectors contribute to this activity, and to understand how effectors responsible for intracellular replication contribute to the induction of gut inflammation. Separately, we observed a similarly strong phenotype for *S*.Tm Zeta, which is deficient for the regulator SpvR and thus impaired for expression of genes on the virulence plasmid regulon *spvABCD*[66,67]. Effectors on this operon reportedly have a range of functions: SpvB is an ADP-ribosyltransferase which causes disruption to the host cytoskeleton and also promotes cell death via apoptosis[68–70]; SpvC has anti-inflammatory functions via its phosphothreonine lyase activity against several MAPK signalling proteins[71–73]; while SpvD acts as a cysteine protease to inhibit NF-κB signalling, possibly by targeting host exportin Xpo2[74,75]. In this study,

we show that *S*.Tm Zeta is strongly attenuated at colonising systemic niches (Fig. 3B) but remains partially competent for inducing SPI-2 T3SS-dependent gut inflammation (Fig. 4C), which suggests these activities are not necessarily linked, but further work is needed to understand how the reported molecular activities of these proteins contributes to these disease phenotypes. In addition, future work should utilise simpler mutants (e.g., single or double mutants) to narrow these multimutant phenotypes to the responsible effectors, followed by complementation of these candidate genes to restore phenotypes to that of the WT parent strain. Ultimately, we recommend this burden of proof when using these multimutant strains (as described in Fig. 5 for *S*.Tm Delta), in order to confidently link mutant genotypes to phenotypes and to discount any confounding impact of polar effects or other genetic interactions.

In conclusion, we describe how multimutants created by sequential deletion of functionally linked genes can be easily used in a variety of experimental contexts to gain new insights into bacterial virulence. We show that effector cohorts linked to intracellular replication and protection from host cell defences are important for migration from the gut to systemic niches, and these same cohorts also contribute to SPI-2 T3SS-dependent gut inflammation. Finally, we show that the effectors SopD2 and GtgE together are necessary for these phenotypes, providing new insights into how the SPI-2 T3SS contributes to gut infection and migration within the host. We anticipate that these multimutants will prove useful in other experimental contexts to provide new insights into bacterial virulence strategies.

## Methods

### Strains used in this study

All bacterial strains used in this study were *S*.Tm SL1344 SB300[76] or derivatives and are listed in Table S2. Strains in cryostorage at −80 °C were streaked to selective media and subsequently used to inoculate overnight cultures comprising lysogeny broth (LB) medium containing appropriate antibiotics (50 μg/ml streptomycin, 50 μg/ml ampicillin, 50 μg/ml kanamycin, or 15 μg/ml chloramphenicol, as required).

### Strain construction

All primers used for strain construction and validation are listed in Table S3. Single mutant strains in an SL1344 background were constructed using the lambda-red protocol, in which a gene of interest is replaced with an antibiotic resistance cassette flanked by FRT sites[22]. Primers were designed with approximately 40 base pairs flanking the gene of interest and 20 base pairs of an antibiotic resistance cassette. Plasmids pKD3 or pKD4 were used as DNA templates in PCR reactions to amplify products suitable for gene replacement with cassettes encoding chloramphenicol (pKD3) or kanamycin (pKD4) resistance via homologous recombination. *S*.Tm SL1344 carrying the plasmid pKD46 was incubated for 3 h at 30 °C in LB containing 50 μg/ml ampicillin and 10 mM arabinose. Cells were washed in ice cold water and concentrated via centrifugation, then transformed with purified DNA via electroporation. Cells were let to recover in LB for 1 h at 37 °C, then plated to LB agar plates containing either 50 μg/ml kanamycin or 15 μg/ml chloramphenicol, as required. Colonies were picked and genotyped via PCR with primers flanking the replaced gene of interest (Table S3). Separately, single mutant strains in 14028S background were generated previously[23]. P22 lysates were generated from these single mutant strains, and used to transfer the deletion of interest to a clean strain of *S*.Tm SL1344, which was subsequently passaged via replating several times to promote clearance of phage and re-genotyped via PCR. Multimutants were similarly constructed by repeated rounds of P22 transduction as above. Strains bearing both chloramphenicol and kanamycin resistance (i.e., after two rounds of P22 transduction) had these resistance cassettes removed via electroporation with pCP20 encoding the *Flp* recombinase flippase. All strains were re-genotyped

after each round of Flp-FRT recombination, to avoid unwanted recombination events at FRT scar sites.

### Chromosomal complementation of effector genes

Mutant strains deficient for *sopD2* and *gtgE* were complemented with these genes via subcloning into a suicide vector followed by conjugation and homologous recombination into recipient strains. Briefly, PCR was used to generate amplicons comprising either *sopD2* or *gtgE* with 1000 bp flanking regions and suitable restriction sites. Amplicons were cloned into vector pSB890 via T4 DNA ligase reactions, then used to transform electrocompetent *E. coli* SM10λpir. Overnight cultures of recipient strains were prepared, then combined with cultures of donor strains. Selection with sucrose and tetracycline was used to identify successful conjugation and recombination events, which were confirmed by genotyping PCR.

### Whole-genome sequencing and bioinformatics analysis

Overnight cultures of multimutant strains were pelleted by centrifugation and genomic DNA was extracted using a QIAmp DNA Mini Kit (Qiagen) according to the manufacturer's instructions. Library preparation and short-read Illumina sequencing were performed by BMKGENE to confirm deletion of target genes and assess the degree of other polymorphisms in the genome. The resulting raw reads were cleaned by removing adaptor sequences, low-quality-end trimming and removal of low-quality reads using BBTools v 38.18 using the parameters $trimq = 14$, $maq = 20$, $maxns = 0$ and $minlength = 45$. (Bushnell, B. BBMap. Available from: https://sourceforge.net/projects/bbmap/.). The genetic changes in the strains as compared to the reference genome (GCF_000210855.2) were identified using breseq (v. 0.38) run in consensus mode with default parameters[77]. Sequencing data is available from the European Nucleotide Archive (ENA) using the accession number PRJEB83585 [ENA Browser].

### Animal husbandry

Animal experiments were conducted in accordance with the Swiss Federal Government guidelines in animal experimentation law (SR 455.163 TVV). Protocols used were approved by the Cantonal Veterinary Office of the canton Zurich, Switzerland (Kantonales Veterinäramt ZH licences 108/2022, 109/2022, 158/2019, 193/2016). Animals were bred and kept under specific pathogen free conditions in individually ventilated cages (light/dark cycle 12:12 hrs, room temperature $21 \pm 1$ °C, humidity $50 \pm 10\%$) (EPIC and RCHCI facilities, ETH Zurich). Wild-type C57BL/6 J mice were used for all in vivo experiments described here. Mice were aged 8-10 weeks at the start of experiments, and a balanced number of males and females was used. Mice were monitored daily and scored for health status in a range of criteria per animal licence requirements, and euthanised prior to experimental endpoint if necessary.

### Animal infection experiments

Mice were infected and treated following the experimental schemes described in each figure and corresponding figure legend. For infections requiring intraperitoneal injection, overnight cultures were incubated on a rotating wheel at 37 °C for 12 h. Overnight cultures were washed in PBS then diluted to achieve ~$10^4$ CFU/ml. For single infections (Fig. 2B), mice received 100 μl of this washed solution by intraperitoneal injection, giving an infectious dose of ~$10^3$ CFU. For mixed infections (Fig. 2D), mice received 100 μl of washed solution to achieve an inoculum of $10^3$ CFU comprising equivalent volumes of each strain, as required. For infections requiring oral gavage (Figs. 3–5), mice were gavaged with 25 mg streptomycin one day prior to infection. Overnight cultures were used to inoculate subcultures which were incubated on a rotating wheel at 37 °C for 4 h. Subcultures were washed in PBS then aliquoted to prepare inocula comprising ~$5 \times 10^7$ CFU in a 50 μl volume, which was delivered to the mice by oral gavage. Faeces

was collected in pre-weighed tubes containing 1 ml PBS and homogenised with a steel ball for 2 min at 25 Hz using a Tissue-Lyser (Qiagen). Mice were euthanised at the indicated time-points, and organs were aseptically removed. CFU per organ was quantified by plating to MacConkey agar containing 50 μg/ml streptomycin. Data for liver, spleen, and mesenteric lymph node is presented as CFU per organ, while data for faeces is presented as CFU per gram of faeces.

## Measurement of genomic barcodes by qPCR

Overnight cultures were inoculated with homogenates of indicated organs to enrich for bacterial genetic material, comprising 100 μl homogenate in 2 ml LB and appropriate antibiotics. Cultures were incubated for 12 h at 37 °C on a rotating wheel. Overnight cultures were pelleted by centrifugation and genomic DNA was extracted using a QIAmp DNA Mini Kit (Qiagen) according to the manufacturer's instructions. The abundance of each genetic tag was measured by qPCR as described previously[78]. Relative proportions of each tag were calculated by dividing the DNA copy number of each tag by the sum of all tags within a sample.

## Tissue culture infection

HeLa cells were maintained in Dulbecco's Modified Eagle Medium (DMEM) containing high glucose, sodium pyruvate, and GlutaMAX™ (ThermoFisher) supplemented with 10% (v/v) foetal bovine serum (ThermoFisher) and 1% (v/v) penicillin-streptomycin (ThermoFisher). Cells at high density were passaged using TrypLE Express (ThermoFisher) as required. At 24 h before infection, HeLa cells were seeded to 12 well plates at an approximate density of $5 \times 10^5$ cells per well. Bacterial strains were incubated overnight at 37 °C, then used to inoculate subcultures which were incubated on a rotating wheel at 37 °C for 4 h. Bacterial subcultures were diluted to an approximate concentration of $2.5 \times 10^7$ cells/ml in DMEM without penicillin-streptomycin. HeLa cells were washed with PBS, then provided with DMEM containing diluted bacteria to achieve an approximate MOI of 50. Plates were briefly centrifuged at $500 \times g$ for 5 min to synchronise and promote infection, then incubated at 37 °C for 30 min. Cells were subsequently washed with PBS then provided with DMEM containing 100 μg/ml gentamicin, and were incubated at 37 °C for 1 h to kill extracellular bacteria. At ~2 h post infection, cells were washed with PBS then lysed with 0.1% Triton X-100 diluted in PBS. Lysed cells were collected via pipette, vortexed briefly, then serially diluted and plated to MacConkey agar containing 50 μg/ml streptomycin. CFU were enumerated after overnight incubation of agar plates at 37 °C.

## Histology

Tissue samples were embedded in O.C.T. (Sakura), snap-frozen in liquid nitrogen, and stored at −80 °C. Cryosections were prepared at 5 μm width and mounted on glass slides, then stained with hematoxylin and eosin (H&E). Pathological evaluation was performed in a blinded manner based on the criteria described previously[37]. Briefly, samples were scored on four criteria: degree of submucosal oedema; infiltration of polymorphonuclear granulocytes into the lamina propria; number of goblet cells; and integrity of the epithelia. Scores for each category were combined to achieve a total score representing the pathological state of each sample.

## Lipocalin-2 ELISA

Homogenised faecal samples were thawed from storage at -20 °C and centrifuged to remove faecal material. Lipocalin-2 levels in the supernatant were quantified using a Lipocalin-2 ELISA kit (R & D Systems) according to the manufacturer's instructions. All samples were analysed in duplicate at three different dilutions (undiluted, 1:20, and 1:400), and concentrations were determined by four parameter logistic regression curve.

## Gentamicin protection assay for caecal tissue

Caecal tissue was aseptically extracted from mice following euthanisation, and incubated for 30 min in PBS containing 400 μg/ml gentamicin to kill extracellular bacteria. Tissue was then washed rigorously six times in PBS, then homogenised with a steel ball for 2 min at 25 Hz using a Tissue-Lyser (Qiagen). CFU was quantified by plating to MacConkey agar containing 50 μg/ml streptomycin.

## Statistical analysis and software

Where applicable, statistical significance was assessed by Mann-Whitney $U$ test, as described in figure legends. No statistical methods were used to predetermine sample sizes. Data collection and analysis were not performed blind to the conditions of the experiments. GraphPad Prism v10 was used to perform statistical tests and generate graphs. BioRender was used to generate some graphical elements, including experimental schemes and Fig. 6. Figures were assembled in Inkscape v1.3.2.

## Reporting summary

Further information on research design is available in the Nature Portfolio Reporting Summary linked to this article.

## Data availability

The whole-genome sequencing data for the twelve multimutant strains generated in this study are available in the European Nucleotide Archive (ENA) under accession number PRJEB83585 [ENA Browser]. Source data are provided with this paper.

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

## Acknowledgements

This work has been funded by grants from the Swiss National Science Foundation (310030B_173338/1, 310030_192567, 10.001.588) to W.-D.H., and also supported by the NCCR Microbiomes, funded by the Swiss National Science Foundation (51NF40_180575; to SS and WDH). J.P.M.N. was supported by a Swiss Government Excellence Scholarship (2019.0843). NMA was funded by the National Institute of Health (AIO83359) and The Welch Foundation (I-1704). The authors would like to thank members of the Hardt lab for productive discussion and for technical support. We would like to thank the staff at RCHCI and EPIC animal facilities, and the staff at the Institute of Microbiology at ETH Zurich for their support.

## Author contributions

J.P.M.N., F.G., P.P., N.B., A.M., S.M., A.S., N.A., S.S., and W.-D.H. conceived and designed the experiments. J.P.M.N., F.G., P.P., N.B., A.M., A.S., M.B., Y.S., S.M., and U.E. performed the experiments and analysed the data. J.P.M.N. and W.-D.H. wrote the manuscript. All authors read, commented, and approved the manuscript.

## Competing interests

The authors declare no competing interests.
