## [Transparent Peer Review file · Nature Communications]

***Salmonella* multimutants enable efficient identification of SPI-2 effector protein function in gut inflammation and systemic colonization**

Corresponding Author: Professor Wolf-Dietrich Hardt

Version 0:

Reviewer comments:

Reviewer #1

(Remarks to the Author)

This manuscript uses a series of multi-mutant *Salmonella* Typhimurium strains to examine contribution to infection in a mouse model system, proposing a system for categorizing effector proteins for multi-mutant generation based on previously established mechanistic information. The research observes differences in colony recovery between wild-type, several multi-mutants and the double mutant strain Δ gtgE Δ sopD2 and observes a difference in caecal histology scoring at one timepoint of interest post-infection (Day 4). The manuscript uses well established protocols and approaches previously implemented for *Salmonella* (ex, multi-mutant studies).

There are some areas of concern, described below, that make interpretation of the data challenging for the reader. In addition, the manuscript text consistently presents an overinterpretation of data beyond what is shown and does not consider other possible explanations for the observations. Finally, the rationale for multi-mutant creation, presented to be a main feature of novelty in the manuscript, is complicated by multiple infection phenotypes for many of the effectors studied. As such, it is difficult to justify for the reader that the combinations chosen are an ideal choice.

Major concerns to be addressed:

- **Multimutant Rationale:** While the overall concept for the selection of genes for multi-mutants could make sense, from a logistical perspective, the strategy of creating “loose groupings” of genes for *Salmonella* is more complicated than described in the manuscript, which makes it difficult to justify for the reader the classification bins. For example, a large proportion *Salmonella* effectors mediate multiple infection phenotypes and as such, they cannot be classified into one group. This is particularly true of SPI-2 effectors. For instance, while the categorization classified SifA and PipB2 as affecting SCV dynamics (Figure 1A), it is also well established that SifA and PipB2 strongly impact cell trafficking. Conversely, SopD2, categorized as affecting cell trafficking, also affects SCV dynamics. Similarly, SifA, PipB2, SopD2 and SseF and SseG also impact immune cell function. As such, the classification of the effectors do not reflect a complete representation based on the literature. There is also concern that some of the categories demonstrate a substantial level of overlap, making classification difficult. This is particularly true of the extremely broad category of “Cell signalling.” Cell signalling is at the heart of countless cellular processes, which overlaps some of the other categories.

- **Strain Construction:** The study uses two different *Salmonella* strains to construct the multi-mutant collection, with some knockouts originating from *S. Typhimurium* 14028S, while the wild-type strain used for all comparisons was *S. Typhimurium* SL1344. While this is not unreasonable if the region in and around the knockout (the entire region to be transferred to SL1344 by phage) is perfectly identical in both strains, this is not the case in this study, as evidenced by Table S1. As a result, the wild-type strain used for all comparisons in this study was SL1344, but the multi-mutants are chimeric strains. The literature demonstrates reports of functional differences in infection between these strains (ex, PMID: 21493681), and we have also observed differences between these two strains in infection, including kinetics. This is an especially important caveat to the interpretation of the experimental results. As the study does not include an approach to confirm that the polymorphisms are not contributing factors to the data, conclusions cannot be made.

- **Complementation:** As standard acceptable practice in bacterial research, all experimental data involving knockouts that

demonstrate a phenotype should also include a parallel comparison with a complementation strain. This is essential to provide support to the reader that the phenotype is not attributed to genetic polymorphisms introduced during the genetic manipulation process or polar effects of the knockout on downstream gene expression. While the manuscript included a complementation strain for the double knockout of *gtgE* and *sopD2* from Figure 5, the majority of the manuscript focuses on the multi-mutants Alpha, Δ invG Alpha, Beta, Δ invG Beta, Gamma, Δ invG Gamma, Delta, Δ invG Delta, Epsilon, Δ invG Epsilon, Zeta and Δ invG Zeta. The data in Figures 2, 3, 4, and Figure S1 suggest the possibility of phenotypes for Alpha, Delta, Zeta, Δ invG Alpha, Δ invG Delta and Δ invG Zeta, but comparisons with the full complementation strains were not performed. Complete complementation strains for Alpha, Delta, Zeta, Δ invG Alpha, Δ invG Delta and Δ invG Zeta should be created and comparison experiments with the associated knockouts should be included. This can be accomplished using the same genomic knockin methodology used for the strains in Figure 5.

- **Multi-mutant Kinetics and Invasion:** Given the polymorphisms reported in Table S1 and concerns about kinetic and functional variations between *S. Typhimurium* SL1344 and 14028S, it is possible that the reported phenotypes are not due to the effectors of interest but rather due growth defects/variations in culture upstream of mouse inoculation that impact infection efficacy. This is especially important to consider with *Salmonella*, as effector expression at the time of inoculation is growth phase dependent (ie, growth impairment can impact the level and nature of effector expression profile in the inoculum, subsequently affecting the progression of infection). The study did not assess the multi-mutants for growth defects or invasion efficiency. It would be essential to establish the absence of growth defects under the same culture conditions used for inoculum preparation as well as invasion defects.

- **Conclusions for Δ gtgE Δ sopD2 and Delta Mutants:** Throughout the manuscript, the text implies that the data indicates that *GtgE* and *SopD2* are important for gut inflammation and mucosal pathology. There are several concerns with this statement. In addition to concerns due to missing experimental controls/comparisons described above and assessments of growth defects and invasion, it is very possible that the effectors do not contribute to gut inflammation at all, and that because the previously established mechanisms occur upstream in the infection life cycle, this suppression of bacterial growth is the underlying reason. Indeed, the paragraph encompassing Line 416-423 and the preceding paragraph highlights the previously established molecular mechanisms for *GtgE* and *SopD2*, and suggest that these previously established roles might contribute to the gut inflammation phenotype. It however does not go far enough to suggest the likely possibility and make it clear that this would mean that they do not control a gut inflammation phenotype. It is also possible that gut inflammation difference is simply a result of delayed infection kinetics with the mutants (not assessed in this manuscript), not that the effectors promote gut inflammation. These possibilities should be highlighted more clearly in the manuscript, and the conclusions should be reworded with this in consideration.

- **Multi-mutants and Analysis of SPI-1 Contribution:** The experiment of multi-mutants with the additional *invG* knockout was missing a parallel comparison with parent multi-mutant strain within the same experiment. Matched comparisons are necessary to make conclusions about relative contribution of SPI-1 vs SPI-2 cohorts.

- **Multi-mutant Clones:** The manuscript reports that two independent clones were created for each multi-mutant and genome sequencing was used for comparison between clones. However given the number of sequence variations between clones, one would have expected that strain comparisons would be included in infection assays. Indeed, in Lines 156-158, the authors indicate "This also provides the opportunity to compare the fitness of each clone, to establish whether particular virulence phenotypes may be attributable to these unwanted polymorphisms." This comparison in mouse would be important to provide some context for the reader, keeping in mind that this would still not address variations common in both clones (ex, regions transferred from 14028S).

- **Strain Sequencing Data:** Is the genome sequencing data deposited fully complete (ex, no gaps or missing regions)? Does this data also include a complete sequencing of all plasmids in each strain? This is also important to ensure that plasmid variations are not responsible for any reported observations. For both, this information should be specified for the reader in the text.

- **Rationale (Line 89-91):** This statement is not reflective on the literature as a whole. One paper cited (PMID 21540636) demonstrates some logistical caveats to the experimental protocol that limit phenotypic observations, as replication was assessed at 6h p.i. for SPI-2 phenotypes, which is not within the conventional range for replication analyses (at best, it may represent early SPI-2 characterized by low levels of effector). The data in other study cited (PMID 34536347) does not reflect the statement in the manuscript. The ability to detect some level of replication defect in many SPI-2 single deletion mutants is consistent with the literature on SPI-2 effectors in nonphagocytic cells and other cell types (ex, PMID: 23592259). As such, it is important not to unintentionally mislead the reader or diminish the value of performing replication assays in culture-based studies. Please reword.

Minor concerns to address:

- Line 25 and 346: The phrase "*Salmonella enterica* spp" does not make sense, as *enterica* is the species. Therefore "spp." should be deleted. Additionally, in Line 24, "*enterica*" should be italicized.
- Line 48: "T3SS" should say "T3SSs"
- Line 65-66: This statement is a too strongly worded given that selected animal model studies have thus far provided some insight. This can be rephrased by replacing "it is not clear" with "it is not fully clear"
- Lines 86-87: This statement reads as vague. Since this is a rationale that sets the stage for this research, please clarify for the reader. If possible, an example would be beneficial.
- Line 93-95: "in many cases, the design of these multimutants precludes...setup" This reads as a bit vague for the reader,

and as such, it is difficult for the reader to gauge if this statement is actually true. Revise with additional information for the reader.

- Figure 1D is redundant with Figure 1C and is overly simplified, as it does not reflect the extent of genetic variation compared to wild-type. If a visual representation is shown, it should include the locations of site variations. This could be shown in a simplistic manner using color coded dots.
- Figure 1D: Why is region between gogA and pipB2 missing for Beta clone 2? This should be also be mentioned in the main text, to be more clear to the reader.
- Figure 1D: Why is epsilon clone 2 missing the region between srfJ and spvR? This should be also be mentioned in the main text, to be more clear to the reader.
- Table S1 may be unintentionally misleading to the reader because it only encompasses small polymorphisms rather than summarizing all genetic variations like insertions and deletions (such as the examples in the following two comments). It would be less confusing to the reader to have

Table S1 encompass all genetic variations from wild-type. In addition, for Table S1, the caption should include all information needed for the reader to understand the table. For example, what do the codes in the "Evidence" column mean? What do the arrows in the "Gene" column mean?

- Lines 133-34: "...with antibiotic resistance cassettes": cite reference
- Lines 135-36: "...existing library of single mutant strains constructed in an S.Tm 14028S background": cite reference
- Line 137-39: cite reference
- Line 145: "unwanted changes" is a lay phrase. Rephrase more scientifically (ex, site variations)
- Line 147: To the reader, the phrase "and are thus suitable for experimental use" is not well suited in this place, given that polymorphisms are subsequently discussed, which do pose a concern to the reader. This phrase can be deleted without taking away from the paragraph.
- Line 151: Replace "are" with "are likely"
- Table 1 is misleading because it does not specify the genetic background of the mutant strains. This could be resolved by adding a column for strain background. For strains that are chimeric (containing regions transferred from 14028S), this should also be specified for the reader
- Line 158: "unwanted" is a lay phrase. Rephrase be more scientifically appropriate.
- Line 158-159: This statement may not be true based on Figure 1D. If so, please revise statement.
- Lines 171-173: Cite the references that show that each of these knockouts lead to lower bacterial loads in the liver and spleen compared to wild-type.
- Line 182: "consistent with previous findings" - cite associated reference
- Lines 189-194: This statement, while possible, is not supported by cited literature. Cite all reference associated with these multi-mutants that support this argument. This statement is also overstated based on the experimental data shown. Please revise.
- Line 246-247: Reference citation is missing.
- Lines 259-260: This is established in the literature for Salmonella. Please include that this is consistent with the literature, and cite references that demonstrate this.
- Lines 340-342: "Collectively, these data provide new mechanistic insight". While survival has been previously linked to mechanism in published literature, this study does not include mechanistic studies and therefore this statement cannot be made. This statement can be revised or omitted.
- Line 349-350: "logistical difficulties...contexts". This statement reads as vague for the reader. More clarification is needed.
- Line 364-368: It is also important to clarify for the reader that a multi-mutant may yield no phenotype simply because of the combination of effectors chosen for the multi-mutant (ex, it is possible that a S. Tm Gamma effector such as gogB may contribute to gut inflammation in concert with another effector not chosen in the multi-mutant). This should be highlighted first, as it is a limitation of this approach.
- Line 461-462: "...are important for migration from the gut to systemic niches, and these same cohorts also contribute to SPI-2 T3SS-dependent gut inflammation." Given the above comments that other possibilities are possible, this is an overstatement. Please rephrase.
- Line 463: "SopD2 and GtgE together are necessary for these phenotypes, providing new insights into how the SPI-2 T3SS contributes to gut infection and migration within the host." Given the above comments that other possibilities are possible, this is an overstatement. Please rephrase.
- Lines 30-31 (Abstract): "we show that three principle phenotypes define the functional contribution of the SPI-2 T3SS to infection." Although the study chooses combinations of effectors to study together based on previously established mechanistic studies, this study is limited in that effector contributions can be missed simply by the choice of the incorrect combination of effectors (refer to comment regarding Line 364-368). As this is not specifically demonstrated by this study, this statement should be revised or omitted.
- Line 38-39 (Abstract): "and thus establish that the SPI-2 effectors SopD2 and GtgE are critical for the promotion of gut inflammation and mucosal pathology" Given the above comments possible scenarios resulting in the observation, this is an overstatement. Please rephrase.
- Methods, Strains used in this study: It should be clarified for the reader that knockouts in 14028S background were also used.
- Methods, Strain construction: It should be specified that for a number of knockouts, strains from a pre-existing collection in 14028S background were used instead, citing the reference for this. It should also be specified which knockouts were used in the 14028S background.
- Lines 487-488: Include a reference to the associated table of primers.

(Remarks to the Author)

In this study Newson et al have developed a set of Salmonella Typhimurium multimutants that they show can be used to investigate the roles of SPI-2 T3SS effector proteins in vivo. The set of 6 multimutants were designed by grouping effectors known to interfere with specific pathogen-host functions, where possible, including intracellular replication and interference with host signal transduction. Alternatively, they were grouped due to gene location (virulence plasmid borne) or into an "unknown function" group. The approach taken to make the multimutants was thoughtful and carefully designed, for example making sure that intergenic sequences were left intact and independently making two of each multimutant to ensure that the results could be replicated. Surprisingly, 3 of the multimutants (representing almost half of the effector repertoire) replicated the WT phenotype in all the tested (murine) models. In my view this is the major weakness of the paper since the approach essentially confirmed that effectors with known functions are important but did not identify any novel effector functions. The paper is well written and was a pleasure to read, and the experimental results were presented clearly and logically. My only writing suggestion is that the results section read more like a combined results/discussion so some of the discussion seemed rather repetitive.

Minor comments

Lines 230-234: This is a little confusing and it could be misunderstood to mean that the results were similar to those seen in the spleen/liver. Maybe clearer to say that the phenotype was not apparent for most strains with only the ssaV deletion mutant and WT Delta having significantly reduced cfu levels.

Line 265-266: The use of the word "while" seems confusing. "nevertheless" would seem more appropriate.

Spelling: There is no Y in Gentamicin.

Figures: Overall, the figures were easy to follow and well organized. With the exception of the bar graphs showing cecal pathology scores, since it was almost impossible to distinguish the four cell types in Figures 3 & 4 (Fig 5E was somewhat better but could still be improved). I would also check the use of color on graphs, which worked very well for me, but might not work for people with color blindness.

Methods

Overall there seems to be good level of detail.

Mouse infections: Presumably the inocula were plated and it would be good to provide the actual cfu instead of stating 10e3 e.g. for each strain in each experiment. How many mice per group in the CI experiment – it looks like 5 but you have stated clearly elsewhere (Fig 2D). Were any of the animal experiments repeated?

Reviewer #3

(Remarks to the Author)

The authors constructed 6 multimutant strains of *S. Typhimurium* with multiple deletions in the cohort of SPI2 T3SS genes grouped according to the similar functions based on the previous studies. This was done to facilitate understanding of the functions of >30 of these genes encoding effector proteins. The mutants were evaluated using various mouse infection models to corroborate previous findings on these genes. Further, the mutants along with other control strains were barcode-tagged to allow simultaneous evaluation of these mutants as a pool in animal infection models.

The study was designed carefully and performed thoroughly with the highest level of rigor, and the results were presented and summarized clearly and professionally with relevant references. Similar approaches (Chen et al 2021. doi.org/10.1016/j.chom.2021.08.012) have been employed and reported to simplify the process of assessing the functions of >30 effector genes. In that sense, the novelty of this study is limited. However, I believe the design principles based on which the genes were divided into 6 cohort groups (thus 6 multimutants) have some distinctive advantage over other similar approaches in elucidating the functions of these effector proteins as exemplified with more detailed characterization of sopD2gtgE double mutants.

I believe this manuscript has a value to be published in Nature Communication. I have a few comments for consideration by the authors to improve the manuscript.

1. lines 157-158: the authors used two independent clones for each multimutant strain to establish if particular phenotypes are attributable to unwanted polymorphisms. I couldn't find any results shown to answer this question.

2. The authors mentioned about context-dependency (line 371) and redundancy of gene functions among the SPI2 T3SS effectors. These concepts are not something unique to these effector proteins but are a general theme in genetics. The more broad and general term for these is "genetic interactions" and redundancy would be the case for negative genetic interactions. Describing these observations in terms of genetic interactions might be a good way to understand them as a more common issues in genetics.

3. Lines 169-173 shows the inclusion of two single deletion mutants (ssaV and efl). If the brief rationale for inclusion of these control groups is provided, it would be helpful for the readers to follow.

4. Line 388. change "reduced" to "attenuated"?

Reviewer #4

(Remarks to the Author)

Version 1:

Reviewer comments:

Reviewer #1

(Remarks to the Author)

The authors have addressed and/or clarified their perspective regarding the majority of comments. A few remaining concerns regarding this feedback are indicated below.

*1. Complementation: As standard acceptable practice in bacterial research, all experimental data involving knockouts that demonstrate a phenotype should also include a parallel comparison with a complementation strain. This is essential to provide support to the reader that the phenotype is not attributed to genetic polymorphisms introduced during the genetic manipulation process or polar effects of the knockout on downstream gene expression. While the manuscript included a complementation strain for the double knockout of *gtgE* and *sopD2* from Figure 5, the majority of the manuscript focuses on the multi-mutants Alpha, Δ invG Alpha, Beta, Δ invG Beta, Gamma, Δ invG Gamma, Delta, Δ invG Delta, Epsilon, Δ invG Epsilon, Zeta and Δ invG Zeta. The data in Figures 2, 3, 4, and Figure S1 suggest the possibility of phenotypes for Alpha, Delta, Zeta, Δ invG Alpha, Δ invG Delta and Δ invG Zeta, but comparisons with the full complementation strains were not performed. Complete complementation strains for Alpha, Delta, Zeta, Δ invG Alpha, Δ invG Delta and Δ invG Zeta should be created and comparison experiments with the associated knockouts should be included. This can be accomplished using the same genomic knockin methodology used for the strains in Figure 5.*

*Response: We appreciate that complementation of deleted genes represents a common practice in bacterial genetics, and indeed represents a fundamental component of the Molecular Koch's postulates of virulence. While this manuscript does provide phenotypic data for all six multimutants and their derivatives, we chose to ultimately focus on *S. Tm* Delta to provide an example of how iterative construction allows for deductively narrowing to the individual effectors responsible for a particular phenotype (Figure 5 of this manuscript describes this process). Indeed, we describe how a double-complemented mutant that is restored for *sopD2* and *gtgE* approximates the WT strain during orogastric infection, satisfying the Molecular postulates in this instance. However, the process to chromosomally complement a multimutant strain is particularly laborious and characterised by low efficiency, and so to perform this complementation for all multimutants and their derivatives would be an enterprise well beyond the scope and requirements of this manuscript (such an exercise would take many months to achieve, and this degree of further genetic manipulation would introduce even more polymorphisms). Indeed, we maintain that the best practice for using these strains is to: 1) identify a phenotype for a particular multimutant; 2) use simpler single/double mutants to identify the responsible effector; then 3) complement this simpler mutant to restore WT-level function. We satisfy these three criteria for one strain of interest as an example of how this approach can be effective, but it remains well beyond the scope of this work to perform this analysis for all strains introduced here. We anticipate future work will take this experimental approach for particular multimutants of interest in different experimental contexts.*

Comment: Thank you for acknowledging the importance of gene complementation. While the knockin approach is a viable strategy, if too laborious, other approaches (ex, shuttle plasmids for complementation) are available and a wide selection of reagents are accessible for this pathogen. While it is important to have a strategy for complementation of knockouts, it remains that the majority of this manuscript focuses on multimutant experiments wherein the effect of knockouts was not validated. This leads to concerns regarding interpretation of the results for the reader. It would be important to add a paragraph to the Discussion regarding other important caveats of this study. In this paragraph, the importance of complementation to rule out the possibility of polar effects on gene expression should be highlighted for the reader, and it should be explained that the majority of data in this manuscript would need to be confirmed by complementation analyses in order to reach conclusions.

*1. Multi-mutants and Analysis of SPI-1 Contribution: The experiment of multi-mutants with the additional *invG* knockout was missing a parallel comparison with parent multi-mutant strain within the same experiment. Matched comparisons are necessary to make conclusions about relative contribution of SPI-1 vs SPI-2 cohorts.*

Response: While interesting to consider, Figure 3 already describes the phenotypes for the parent multimutants at a cost of ~45 animals, while Figure 4 describes that of the Δ invG derivatives using another ~50 animals. Further animal use to directly compare Δ invG derivatives to parent strains would not seem to be a prudent use of animal resources, and in our opinion the existing data describes these differences comprehensively and sufficiently.

Comment: This explanation is admirable from a cost perspective, however matched comparisons within the same experiment are needed to draw conclusions from the results. This caveat should also be highlighted in the Discussion for the reader.

Figure 1D: Why is region between gogA and pipB2 missing for Beta clone 2? This should be also be mentioned in the main text, to be more clear to the reader.

Response: Due to the random nature of genomic packaging by the P22 phage, there is some variability in which region of the 14028 chromosome surrounding the deleted gene is transferred to the recipient SL1344 strain. The sequencing data for S.Tm Beta clones 1 and 2 provide a particularly striking example of this. During strain construction, gogA::cat was transferred from a 14028S donor strain to the SL1344 recipient, which resulted in different regions of the 14028S chromosome surrounding the deletion being packaged and incorporated between both recipients (i.e. clone 1 and clone 2 of the intended mutant). Our whole genome sequencing analysis uses an algorithm that can call regions of high polymorphism as deletions, which produces these large areas of reads that fail to map correctly to the SL1344 genome (visually represented as large blue bars shown for S.Tm Beta clone 2 but not for S.Tm Beta clone 1, on the left side of Fig 1D). Thus, this region is not missing for this clone, it is simply a graphical artefact of a high degree of variation between the SL1344 and 14028S chromosomes in this genetic location.

Comment: Thank you for the additional information. This information would be important for the reader to understand the data. This can be addressed by adding an asterisk in the figure and adding an associated note in figure caption to explain.

Figure 1D: Why is epsilon clone 2 missing the region between srfJ and spvR? This should be also be mentioned in the main text, to be more clear to the reader.

Response: As above, this is a region of high variation between the SL1344 and 14028S chromosomes that results in the algorithm tentatively flagging this region as a deletion.

Comment: Please see response above regarding this clone.

Reviewer #3

(Remarks to the Author)

The authors addressed all concerns I raised in the initial review sufficiently and thoroughly. I also reviewed the critiques of the other reviewers and the responses by the authors, which gave me an additional level of confidence that the revision process improved the manuscript significantly. I recommend this manuscript for publication.

Reviewer #4

(Remarks to the Author)

Open Access This Peer Review File is licensed under a Creative Commons Attribution 4.0 International License, which permits use, sharing, adaptation, distribution and reproduction in any medium or format, as long as you give appropriate credit to the original author(s) and the source, provide a link to the Creative Commons license, and indicate if changes were

made.

Point-by-point response to the reviewers' comments:

Reviewer #1:

This manuscript uses a series of multi-mutant Salmonella Typhimurium strains to examine contribution to infection in a mouse model system, proposing a system for categorizing effector proteins for multi-mutant generation based on previously established mechanistic information. The research observes differences in colony recovery between wild-type, several multi-mutants and the double mutant strain Δ gtgE Δ sopD2 and observes a difference in caecal histology scoring at one timepoint of interest post-infection (Day 4). The manuscript uses well established protocols and approaches previously implemented for Salmonella (ex, multi-mutant studies).

There are some areas of concern, described below, that make interpretation of the data challenging for the reader. In addition, the manuscript text consistently presents an overinterpretation of data beyond what is shown and does not consider other possible explanations for the observations. Finally, the rationale for multi-mutant creation, presented to be a main feature of novelty in the manuscript, is complicated by multiple infection phenotypes for many of the effectors studied. As such, it is difficult to justify for the reader that the combinations chosen are an ideal choice.

Response: We thank the Reviewer for the constructive feedback and are grateful for the in-depth examination of the manuscript's strengths and weaknesses. We hope the revised version allays concerns regarding the design, interpretation, and experimental use of these strains, and we maintain that these mutants represent a useful toolset for interrogating effector function in various experimental contexts.

Major concerns to be addressed:

- 1. Multimutant Rationale: While the overall concept for the selection of genes for multi-mutants could make sense, from a logistical perspective, the strategy of creating "loose groupings" of genes for Salmonella is more complicated than described in the manuscript, which makes it difficult to justify for the reader the classification bins. For example, a large proportion Salmonella effectors mediate multiple infection phenotypes and as such, they cannot be classified into one group. This is particularly true of SPI-2 effectors. For instance, while the categorization classified SifA and PipB2 as affecting SCV dynamics (Figure 1A), it is also well established that SifA and PipB2 strongly impact cell trafficking. Conversely, SopD2, categorized as affecting cell trafficking, also affects SCV dynamics. Similarly, SifA, PipB2, SopD2 and SseF and SseG also impact immune cell function. As such, the classification of the effectors do not reflect a complete representation based on the literature. There is also concern that some of the categories demonstrate a substantial level of overlap, making classification difficult. This is particularly true of the extremely broad category of "Cell signalling." Cell signalling is at the heart of countless cellular processes, which overlaps some of the other categories.*

Response: This is an important first comment, and we appreciate the chance to clarify our rationale. The loose grouping of effectors based on reported literature is simply a semi-guided attempt to create rationally designed mutants that should produce strong phenotypes, rather than randomly deleting effectors together with no regard to their function. In previous approaches of this kind (e.g. PMID: 31235639, 34536347), using a "random" sequence of deletions resulted sooner or later in the deletion of critical effectors which strongly attenuated that strain and all subsequent mutants. This may obscure phenotypic contributions of

additionally deleted genes. We aimed to avoid such bias, and we maintain that the elimination of multiple effectors in the S.Tm Beta, Gamma, and Epsilon mutants provide important information on the redundancy of the respective effectors, and provide useful starting points for further deletion of additional effectors to study their *in vivo* functions in future studies.

Having said this, we agree that it is not possible to neatly categorize many effectors into a definitive functional group, and the Reviewer well describes how some effectors reportedly contribute to more than one such group. Similarly, 'cell signalling' is given as an umbrella term for various aspects of innate immune signalling (including but not limited to manipulation of NF- κ B, TNF, and MAPK signalling). We do not and cannot capture the full and accurate current state of the literature with one introductory panel in one figure, and we defer to very well-regarded Reviews that effectively summarise current literature (cited in the Introduction and Discussion). Our purpose with this first panel is simply to introduce a general audience to the concepts of functional grouping, which underpin the rational design of mutants described in **Fig 1C**. Without such an introductory overview, the reader is left to question how genes might be rationally grouped. We describe also in the first Results section our criteria for grouping effector deletions together, and we hope that the reader is left with the impression that such groupings and deletions produce an acceptable version of multimutant strains that are useful for iteratively and deductively screening for effector phenotypes, an approach which will likely help to overcome current challenges in the field.

- 2. Strain Construction: The study uses two different Salmonella strains to construct the multi-mutant collection, with some knockouts originating from S. Typhimurium 14028S, while the wild-type strain used for all comparisons was S. Typhimurium SL1344. While this is not unreasonable if the region in and around the knockout (the entire region to be transferred to SL1344 by phage) is perfectly identical in both strains, this is not the case in this study, as evidenced by Table S1. As a result, the wild-type strain used for all comparisons in this study was SL1344, but the multi-mutants are chimeric strains. The literature demonstrates reports of functional differences in infection between these strains (ex, PMID: 21493681), and we have also observed differences between these two strains in infection, including kinetics. This is an especially important caveat to the interpretation of the experimental results. As the study does not include an approach to confirm that the polymorphisms are not contributing factors to the data, conclusions cannot be made.*

Response: This is an interesting comment, and we appreciate the Reviewer for raising this concern. We acknowledge that SL1344 and 14028S do have important differences, and our group has published work describing some examples in the past (PMID: 38460128, 37348498). Indeed, the current manuscript leverages an established library of single mutants constructed in a 14028S background (PMID: 25007190), followed by transduction into a clean SL1344 strain. It is true that generalised transduction followed by plating to selective media generally results in the transfer of a region of several thousand base pairs from the donor chromosome to that of the recipient, and the region packaged by the P22 phage is random (thus, individual transduction recipients will receive and incorporate slightly different regions of the donor chromosome, so long as the resistance cassette is incorporated). There are several reasons to explain why this is considered acceptable practice. First, the construction and use of two independently-constructed clones (which have incorporated slightly different elements of donor DNA) allows confidence that the phenotypes observed during experiments are not artefacts of chimeric strain construction. In the revised manuscript, we experimentally test both independently made strains for all multimutants, and observe very consistent phenotypes for both clones of each mutant (this is presented in **Fig**

S1B, **Fig S2C**, and **Fig S3B**, and is more fully addressed in the corresponding Reviewer comment below). Further, it is a well-established practice to transduce single deletions from 14028S into a SL1344 background, and indeed our group has successfully leveraged this strategy in numerous publications over the last decade (PMID: 39966379, 32416061, 40315408, among others). Similarly, other groups have utilized transfer of single deletions from 14028S background to SL1344 (PMID 33139383, 33436434, 34460873), and so we argue that this is acceptable practice for the field, especially when supported by whole genome sequencing data that permits identification of which regions of donor DNA have been integrated into a transduction recipient. In this manuscript, we provide genome sequencing analysis (**Fig 1D**, **Table S2**) that reports and describes regions or polymorphisms transduced from 14028, which provides a powerful level of transparency regarding strain construction, beyond that which is typically reported in the field. Ultimately, we argue that transduction of small regions of DNA from 14028 to SL1344 is acceptable practice, both here and for the broader field. In the revised version of the manuscript, we have pointed out this caveat in the Discussion section. Note: **Table S1** in the original manuscript is now **Table S2** in the revised manuscript.

- 3. Complementation: As standard acceptable practice in bacterial research, all experimental data involving knockouts that demonstrate a phenotype should also include a parallel comparison with a complementation strain. This is essential to provide support to the reader that the phenotype is not attributed to genetic polymorphisms introduced during the genetic manipulation process or polar effects of the knockout on downstream gene expression. While the manuscript included a complementation strain for the double knockout of *gtgE* and *sopD2* from Figure 5, the majority of the manuscript focuses on the multi-mutants Alpha, Δ invG Alpha, Beta, Δ invG Beta, Gamma, Δ invG Gamma, Delta, Δ invG Delta, Epsilon, Δ invG Epsilon, Zeta and Δ invG Zeta. The data in Figures 2, 3, 4, and Figure S1 suggest the possibility of phenotypes for Alpha, Delta, Zeta, Δ invG Alpha, Δ invG Delta and Δ invG Zeta, but comparisons with the full complementation strains were not performed. Complete complementation strains for Alpha, Delta, Zeta, Δ invG Alpha, Δ invG Delta and Δ invG Zeta should be created and comparison experiments with the associated knockouts should be included. This can be accomplished using the same genomic knockin methodology used for the strains in Figure 5.*

Response: We appreciate that complementation of deleted genes represents a common practice in bacterial genetics, and indeed represents a fundamental component of the Molecular Koch's postulates of virulence. While this manuscript does provide phenotypic data for all six multimutants and their derivatives, we chose to ultimately focus on *S.Tm* Delta to provide an example of how iterative construction allows for deductively narrowing to the individual effectors responsible for a particular phenotype (Figure 5 of this manuscript describes this process). Indeed, we describe how a double-complemented mutant that is restored for *sopD2* and *gtgE* approximates the WT strain during orogastric infection, satisfying the Molecular postulates in this instance. However, the process to chromosomally complement a multimutant strain is particularly laborious and characterised by low efficiency, and so to perform this complementation for all multimutants and their derivatives would be an enterprise well beyond the scope and requirements of this manuscript (such an exercise would take many months to achieve, and this degree of further genetic manipulation would introduce even more polymorphisms). Indeed, we maintain that the best practice for using these strains is to: 1) identify a phenotype for a particular multimutant; 2) use simpler single/double mutants to identify the responsible effector; then 3) complement this simpler mutant to restore WT-level function. We satisfy these three criteria for one strain of interest

as an example of how this approach can be effective, but it remains well beyond the scope of this work to perform this analysis for all strains introduced here. We anticipate future work will take this experimental approach for particular multimutants of interest in different experimental contexts.

4. *Multi-mutant Kinetics and Invasion: Given the polymorphisms reported in Table S1 and concerns about kinetic and functional variations between S. Typhimurium SL1344 and 14028S, it is possible that the reported phenotypes are not due to the effectors of interest but rather due growth defects/variations in culture upstream of mouse inoculation that impact infection efficacy. This is especially important to consider with Salmonella, as effector expression at the time of inoculation is growth phase dependent (ie, growth impairment can impact the level and nature of effector expression profile in the inoculum, subsequently affecting the progression of infection). The study did not assess the multi-mutants for growth defects or invasion efficiency. It would be essential to establish the absence of growth defects under the same culture conditions used for inoculum preparation as well as invasion defects.*

Response: This is an important consideration, and we thank the Reviewer for this comment. Indeed, improper strain construction could introduce differences between these strains that would impact strain phenotypes *in vivo*. To assess invasion defects, we measured invasion *in vitro* using the well-established gentamicin protection assay (presented in the revised manuscript as **Fig. S3B**). At approximately 2 hours post infection of HeLa cells, we observed roughly equivalent CFU for WT and two independent clones of all multimutants ($\sim 10^6$ CFU/ml), while we recovered much fewer CFU for the invasion deficient *S.Tm* $\Delta invG$ ($\sim 10^3$ CFU/ml). These data suggest no invasion deficiency for any of our constructed strains.

To address potential differences in growth and how this might affect strain performance *in vivo*, we first provide CFU data for the inocula used for all animal experiments involving these strains (presented in the revised manuscript as **Fig. S1A** right panel, **Fig. S3A**, and **Fig. S4C**). In all cases, we observe no differences between these strains after the appropriate culture conditions (12 hours at 37 °C for intraperitoneal injection, 4 hours at 37 °C for orogastric infection). Thus, differences observed in animal experiments (**Fig. 2-4**) only arise during infection. To further support these observations, we prepared a series of mock inocula grown under either of these growth conditions, and assessed bacterial growth by CFU plating as above. These data are presented below:

CFU data for these mock inocula are consistent with those observed when inocula used for experiments, as shown in **Fig. S1A**, **Fig. S3A**, and **Fig. S4C**. Together, these data suggest there is no appreciable difference in viability when these strains are grown on selective media. This is consistent with the whole genome sequencing data presented in **Fig. 1D** and **Table S2** which show minimal genetic variation relative to the WT parent strain. Thus we argue that the multimutant phenotypes described here arise from genuine attenuation of these strains due to deletion of various effector genes, and not from artefacts or errors arising from strain construction.

5. *Conclusions for Δ gtgE Δ sopD2 and Delta Mutants: Throughout the manuscript, the text implies that the data indicates that GtgE and SopD2 are important for gut inflammation and mucosal pathology. There are several concerns with this statement. In addition to concerns due to missing experimental controls/comparisons described above and assessments of growth defects and invasion, it is very possible that the effectors do not contribute to gut inflammation at all, and that because the previously established mechanisms occur upstream in the infection life cycle, this suppression of bacterial growth is the underlying reason. Indeed, the paragraph encompassing Line 416-423 and the preceding paragraph highlights the previously established molecular mechanisms for GtgE and SopD2, and suggest that these previously established roles might contribute to the gut inflammation phenotype. It however does not go far enough to suggest the likely possibility and make it clear that this would mean that they do not control a gut inflammation phenotype. It is also possible that gut inflammation difference is simply a result of delayed infection kinetics with the mutants (not assessed in this manuscript), not that the effectors promote gut inflammation. These possibilities should be highlighted more clearly in the manuscript, and the conclusions should be reworded with this in consideration.*

Response: The central criticism in this comment appears to be the link between bacterial numbers and a corresponding impact on gut inflammation (ie the S.Tm Delta and Δ sopD2gtgE mutants produced impaired gut pathology because there are relatively few surviving bacteria). There are several points to contend here. Firstly, while previous work does thoroughly characterise the molecular mechanisms of these effectors, much of the phenotypic work *in vivo* has been done using intraperitoneal infection, and so there is little to no information regarding their importance to gut infection and gut mucosa inflammation. Such data is a key part of our manuscript, and so our work extends previous work in an important manner. Secondly, we report in various instances that S.Tm Delta and its derivatives are still present to appreciable levels in the gut: **Fig 3C** shows only a modest reduction in CFU in the mesenteric lymph node, **Fig 3D** shows approximately equal carriage in the faeces (taken as a proxy of gut luminal carriage), while intracellular populations within caecal tissue are reduced but not eliminated (**Fig 5D**). We tend to agree that the pathogen's tissue loads are likely one critical parameter which determines if (or when) a mutant will elicit tissue-destructive inflammatory pathology. However, previous reports show that the intensity of gut tissue pathology has a non-linear relationship to gut tissue loads (PMID: 18268033, 15661931). Therefore, we agree it is possible that slight kinetic delays or reductions of absolute pathogen densities in the gut tissue may explain why the S.Tm Delta and Δ sopD2gtgE mutants elicit a much weaker gut inflammation response, relative to the WT and Δ invG strains. However, our *in vivo* mouse infection data cannot exclude the alternative possibility that SopD2 and/or GtgE may trigger specific pro-inflammatory signalling responses in some cells of the gut. We cite relevant literature to describe the known activity of these effectors, and it is tempting to speculate how such molecular activity might translate into gut inflammation phenotypes, but we purposefully reserve further investigation at the

molecular and cellular level for future work, given the already broad scope of this manuscript. Overall, the revised manuscript more clearly approaches these possibilities in the Discussion section.

- 6. Multi-mutants and Analysis of SPI-1 Contribution: The experiment of multi-mutants with the additional invG knockout was missing a parallel comparison with parent multi-mutant strain within the same experiment. Matched comparisons are necessary to make conclusions about relative contribution of SPI-1 vs SPI-2 cohorts.*

Response: While interesting to consider, **Figure 3** already describes the phenotypes for the parent multimutants at a cost of ~45 animals, while **Figure 4** describes that of the $\Delta invG$ derivatives using another ~50 animals. Further animal use to directly compare $\Delta invG$ derivatives to parent strains would not seem to be a prudent use of animal resources, and in our opinion the existing data describes these differences comprehensively and sufficiently.

- 7. Multi-mutant Clones: The manuscript reports that two independent clones were created for each multi-mutant and genome sequencing was used for comparison between clones. However given the number of sequence variations between clones, one would have expected that strain comparisons would be included in infection assays. Indeed, in Lines 156-158, the authors indicate "This also provides the opportunity to compare the fitness of each clone, to establish whether particular virulence phenotypes may be attributable to these unwanted polymorphisms." This comparison in mouse would be important to provide some context for the reader, keeping in mind that this would still not address variations common in both clones (ex, regions transferred from 14028S).*

Response: We thank the Reviewer for raising this important point. Indeed, the original manuscript describes the construction and validation of two independently constructed clones, but only one clone of each is experimentally characterised. In the revised version of this manuscript, we expand our analysis to compare both clones of each multimutant strain. During intraperitoneal infection, we find a broadly similar phenotype for the second clone compared to the first, both during single infection (**Fig S1B**) and during competitive index infections (**Fig S2C**). We did not extend this analysis to oral infection due to the large number of additional animals required, but in the main figures we see generally similar phenotypes for the first clones when comparing intraperitoneal and orogastric infections (**Fig 2** vs **Fig 3**), so we anticipate the second clones also behave similarly to the first clones in this model too.

- 8. Strain Sequencing Data: Is the genome sequencing data deposited fully complete (ex, no gaps or missing regions)? Does this data also include a complete sequencing of all plasmids in each strain? This is also important to ensure that plasmid variations are not responsible for any reported observations. For both, this information should be specified for the reader in the text.*

Response: Our analysis shows no deletions, rearrangements or other genetic changes other than those specifically reported in **Fig 1D** and **Table S2** (i.e. genes that have been correctly deleted are absent, polymorphisms relevant to the reference genome are reported, no additional elements such as phages have been mistakenly incorporated). Complete sequencing of the three SL1344-associated plasmids is also provided in the deposited resource, and **Fig 1D** similarly describes changes to these regions (notably, the plasmid-

encoded *spvR* is correctly shown to be absent for *S.Tm Zeta*). These data are publicly available for review at ENA: PRJEB83585, as described in the Methods section.

9. *Rationale (Line 89-91): This statement is not reflective on the literature as a whole. One paper cited (PMID 21540636) demonstrates some logistical caveats to the experimental protocol that limit phenotypic observations, as replication was assessed at 6h p.i. for SPI-2 phenotypes, which is not within the conventional range for replication analyses (at best, it may represent early SPI-2 characterized by low levels of effector). The data in other study cited (PMID 34536347) does not reflect the statement in the manuscript. The ability to detect some level of replication defect in many SPI-2 single deletion mutants is consistent with the literature on SPI-2 effectors in nonphagocytic cells and other cell types (ex, PMID: 23592259). As such, it is important not to unintendedly mislead the reader or diminish the value of performing replication assays in culture-based studies. Please reword.*

Response: The first citation (PMID 21540636) assesses replication of SPI-2 mutants in both epithelial cells (6 hours post infection) and RAW26.47 cells (24 hours post infection), which does allow a suitable timeframe for measuring replication, thus this citation does correctly support our comment that many SPI-2 effector single mutants show no replication phenotype *in vitro*. Similarly, the second citation (PMID 34536347) does indeed support this same comment, particularly Figure S1 in this citation which extensively characterises a range of single mutants in both epithelial cells and macrophages at 18 hours post infection (certainly a replication-relevant timepoint). We appreciate the additional provision of citation PMID: 23592259 (we have added this to the revised manuscript), but this too also demonstrates that single deletions for many effectors do not result in an appreciable replication phenotype, which is consistent with our comments in this section. Our central argument is that many effectors show little to no phenotype as single deletions, and that the use of such mutants is laborious and particularly not suited to experiments that require higher throughput, costly resources, or ethical considerations (e.g. animal lives). Despite these concerns, we do already provide several citations for instances where single mutants have been used effectively, so we feel this still represents a balanced view of the literature and of available tools. Thus, we maintain our comments here are suitable as written.

Minor concerns to address:

Line 25 and 346: The phrase “Salmonella enterica spp” does not make sense, as enterica is the species. Therefore “spp.” should be deleted. Additionally, in Line 24, “enterica” should be italicized.

Response: We have made the suggested changes.

Line 48: “T3SS” should say “T3SSs”

Response: This has been corrected.

Line 65-66: This statement is a too strongly worded given that selected animal model studies have thus far provided some insight. This can be rephrased by replacing “it is not clear” with “it is not fully clear”

Response: We have made the suggested change.

Lines 86-87: This statement reads as vague. Since this is a rationale that sets the stage for this research, please clarify for the reader. If possible, an example would be beneficial.

Response: Some additional examples have been added to better illustrate the point.

Line 93-95: "in many cases, the design of these multimutants precludes...setup" This reads as a bit vague for the reader, and as such, it is difficult for the reader to gauge if this statement is actually true. Revise with additional information for the reader.

Response: This sentence has been expanded to provide a key limitation of prior work, and better highlights the advantages of the current approach.

Figure 1D is redundant with Figure 1C and is overly simplified, as it does not reflect the extent of genetic variation compared to wild-type. If a visual representation is shown, it should include the locations of site variations. This could be shown in a simplistic manner using color coded dots.

Response: **Figures 1C** and **1D** serve distinct purposes in this manuscript. **Fig 1C** shows the final intended genotype of each of the six multimutants (middle) and the sequence in which corresponding effectors were deleted (right). This information is important for subsequent efforts that rely on deductively narrowing candidate effectors to phenotypes, as demonstrated in **Fig 5**. In **Fig 1D**, we present a graphical representation of regions where sequencing reads fail to map to the SL1344 chromosome (and thus these genes have been deleted). This data is important to demonstrate to the reader that each effector has been correctly deleted from both clones of the intended mutant, while other effectors remain intact. This is especially important given that P22 transduction can inadvertently delete or restore neighbouring effectors, and we show here that this has been carefully avoided due to strain design and validation. We do appreciate the Reviewer's point that polymorphisms arising from strain construction are an important consideration for the reader, and we provide a comprehensive list of such genetic changes in the supplementary material. Graphically presenting single nucleotide polymorphisms for twelve strains across the entire SL1344 chromosome and three plasmids is extraordinarily challenging from a design and logistics viewpoint, and so we feel the current data shown in Fig 1D provides a more convincing and accessible representation of the validity of these strains.

Figure 1D: Why is region between gogA and pipB2 missing for Beta clone 2? This should be also be mentioned in the main text, to be more clear to the reader.

Response: Due to the random nature of genomic packaging by the P22 phage, there is some variability in which region of the 14028 chromosome surrounding the deleted gene is transferred to the recipient SL1344 strain. The sequencing data for S.Tm Beta clones 1 and 2 provide a particularly striking example of this. During strain construction, *gogA::cat* was transferred from a 14028S donor strain to the SL1344 recipient, which resulted in different regions of the 14028S chromosome surrounding the deletion being packaged and incorporated between both recipients (i.e. clone 1 and clone 2 of the intended mutant). Our whole genome sequencing analysis uses an algorithm that can call regions of high polymorphism as deletions, which produces these large areas of reads that fail to map correctly to the SL1344 genome (visually represented as large blue bars shown for S.Tm

Beta clone 2 but not for S.Tm Beta clone 1, on the left side of Fig 1D). Thus, this region is not missing for this clone, it is simply a graphical artefact of a high degree of variation between the SL1344 and 14028S chromosomes in this genetic location.

Figure 1D: Why is epsilon clone 2 missing the region between srfJ and spvR? This should be also be mentioned in the main text, to be more clear to the reader.

Response: As above, this is a region of high variation between the SL1344 and 14028S chromosomes that results in the algorithm tentatively flagging this region as a deletion.

Table S1 may be unintentionally misleading to the reader because it only encompasses small polymorphisms rather than summarizing all genetic variations like insertions and deletions (such as the examples in the following two comments). It would be less confusing to the reader to have Table S1 encompass all genetic variations from wild-type. In addition, for Table S1, the caption should include all information needed for the reader to understand the table. For example, what do the codes in the “Evidence” column mean? What do the arrows in the “Gene” column mean?

Response: We thank the Reviewer for raising these concerns. Our whole genome sequencing analysis did not detect any small insertions or deletions beyond what is reported already in **Fig. 1D**. Separate analysis for phage detection observed the same number of phages in the reference genome and in the re-sequenced multimutants. Regarding the supplementary table, in the revised manuscript, we include a Table title which clarifies some of these details (Table S1 in the original manuscript is now Table S2 in the revised version). To restate here, the evidence column summarises how a polymorphism is called per the breseq pipeline, typically either read alignment (RA) or new junction (JC) (further details are provided in the breseq documentation at <https://bioweb.pasteur.fr/docs/modules/breseq/0.23/output.html>). The arrows in the ‘gene’ column describe which chromosomal strand carries the indicated gene (left arrow denotes negative strand, right arrow denotes positive strand). We hope the provision of a Table legend enhances the accessibility of this supplemental material.

Lines 133-34: “...with antibiotic resistance cassettes”: cite reference

Response: A citation has been added for the Datsenko *et al* (2000) paper that describes deletion of genes by lambda red recombination.

Lines 135-36: “...existing library of single mutant strains constructed in an S.Tm 14028S background”: cite reference

Response: A citation has been added for the Porwollik *et al* (2014) paper that reports construction of the single mutant library in S.Tm 14028S background.

Line 137-39: cite reference

Response: A citation has been added for the Cherepanov and Wackernagel (1995) paper describing Flp-mediated excision of resistance cassettes from the bacterial chromosome.

Further, references have been added for some of the original P22 papers from Zinder and Lederberg.

Line 145: “unwanted changes” is a lay phrase. Rephrase more scientifically (ex, site variations)

Response: We have made the suggested change.

Line 147: To the reader, the phrase “and are thus suitable for experimental use” is not well suited in this place, given that polymorphisms are subsequently discussed, which do pose a concern to the reader. This phrase can be deleted without taking away from the paragraph.

Response: Concerns regarding strain construction and validation, while well-founded, are addressed elsewhere in this response letter and in the revised manuscript.

Line 151: Replace “are” with “are likely”

Response: We have made the recommended change.

Table 1 is misleading because it does not specify the genetic background of the mutant strains. This could be resolved by adding a column for strain background. For strains that are chimeric (containing regions transferred from 14028S), this should also be specified for the reader

Response: This is indeed an important level of detail that we are pleased to include in the revised manuscript. A new table has been added to the supplementary material (**Table S1**) that describes details regarding strain construction, highlighting in particular the genetic background in which each single mutant was constructed (i.e. SL1344 or 14028S). We anticipate that this new material, in combination with **Figure 1D** and **Table S2** will provide a comprehensive level of detail regarding strain construction and validation.

Line 158: “unwanted” is a lay phrase. Rephrase be more scientifically appropriate.

Response: This word has been deleted, the meaning of the sentence is unchanged.

Line 158-159: This statement may not be true based on Figure 1D. If so, please revise statement.

Response: This statement is indeed supported by our whole genome sequencing data, as we detect only the deletions reported in **Fig 1D** and the polymorphisms reported in **Table S2**. Variations between clones that are graphically displayed in **Fig 1D** are addressed in this response letter above.

Lines 171-173: Cite the references that show that each of these knockouts lead to lower bacterial loads in the liver and spleen compared to wild-type.

Response: Added citations to Shea et al (1999) and Chen et al (2021) that respectively show strong attenuation during systemic infection for S.Tm that are SPI-2 structurally-deficient and for S.Tm that are SPI-2 effectorless.

Line 182: “consistent with previous findings” - cite associated reference

Response: Added citation to Newson et al (2025) that shows delayed gut colonisation for a SPI-2 mutant in a systemic infection model.

Lines 189-194: This statement, while possible, is not supported by cited literature. Cite all reference associated with these multi-mutants that support this argument. This statement is also overstated based on the experimental data shown. Please revise.

Response: This statement has been deleted from the manuscript, in response to this comment and another by Reviewer 2.

Line 246-247: Reference citation is missing.

Response: Added citations to Barthel *et al* (2003), Hapfelmeier *et al* (2005), and Stecher *et al* (2005), which together demonstrate the contribution of the SPI-1 T3SS to gut inflammation and enteropathy during orogastric infection of mice.

Lines 259-260: This is established in the literature for Salmonella. Please include that this is consistent with the literature, and cite references that demonstrate this.

Response: Added citations to Hapfelmeier *et al* (2005), Coburn *et al* (2005), and Coombes *et al* (2005), which establish the importance of the SPI-2 T3SS in colonising systemic niches during orogastric infection.

Lines 340-342: “Collectively, these data provide new mechanistic insight”. While survival has been previously linked to mechanism in published literature, this study does not include mechanistic studies and therefore this statement cannot be made. This statement can be revised or omitted.

Response: This statement in the Results has been revised to omit comments regarding mechanism, and such speculation is reserved for the Discussion.

Line 349-350: “logistical difficulties...contexts”. This statement reads as vague for the reader. More clarification is needed.

Response: Examples of the limitations of using single deletion mutants are described in the Introduction. We are inclined to maintain this comment as written to avoid repetition and maintain brevity in the opening sentence of the Discussion.

Line 364-368: It is also important to clarify for the reader that a multi-mutant may yield no phenotype simply because of the combination of effectors chosen for the multi-mutant (ex, it

is possible that a S. Tm Gamma effector such as gogB may contribute to gut inflammation in concert with another effector not chosen in the multi-mutant). This should be highlighted first, as it is a limitation of this approach.

Response: This is indeed an interesting possibility (and grounds for future research), and we have added a sentence to this section to include this explanation.

Line 461-462: "...are important for migration from the gut to systemic niches, and these same cohorts also contribute to SPI-2 T3SS-dependent gut inflammation." Given the above comments that other possibilities are possible, this is an overstatement. Please rephrase.

Response: We have revised our discussion of the mechanistic possibilities regarding how SopD2 and GtgE contribute to gut inflammation, in response to Reviewer 1 major comment #5 (above). Having addressed this, our conclusion that cohorts that are deficient for migration are also deficient for gut inflammation remains sound and supported by the data (though we carefully do not imply mechanistic links beyond that already discussed for the Δ sopD2gtgE mutant). We therefore would maintain this statement as written.

Line 463: "SopD2 and GtgE together are necessary for these phenotypes, providing new insights into how the SPI-2 T3SS contributes to gut infection and migration within the host." Given the above comments that other possibilities are possible, this is an overstatement. Please rephrase.

Response: As above, while we do not distinguish between inflammation that is linked to pathogen load or inflammation that is linked to manipulation of inflammatory signalling, these statements remain sound as written.

Lines 30-31 (Abstract): "we show that three principle phenotypes define the functional contribution of the SPI-2 T3SS to infection." Although the study chooses combinations of effectors to study together based on previously established mechanistic studies, this study is limited in that effector contributions can be missed simply by the choice of the incorrect combination of effectors (refer to comment regarding Line 364-368). As this is not specifically demonstrated by this study, this statement should be revised or omitted.

Response: This comment is addressed above, and the revised Discussion raises this possibility. It is also possible that there are indeed only three principle phenotypes discoverable by deletion of SPI-2 effectors, described here and previously. Our statement in the abstract remains representative of the data described in this manuscript, and it is not feasible to address all hypothetical limitations within the scope of an abstract.

Line 38-39 (Abstract): "and thus establish that the SPI-2 effectors SopD2 and GtgE are critical for the promotion of gut inflammation and mucosal pathology" Given the above comments possible scenarios resulting in the observation, this is an overstatement. Please rephrase.

Response: This is again addressed in previous comments. By either possibility discussed in this manuscript and revision letter, the phenotype we describe remains clear and well supported by the data.

Methods, Strains used in this study: It should be clarified for the reader that knockouts in 14028S background were also used.

Response: This is an important detail and we thank the Reviewer for the opportunity to clarify our strain construction process. In the revised manuscript, additional details regarding SL1344 and 14028S backgrounds for single mutants are described in the Methods and clarified in a new supplementary **Table S1**. Additionally, the revised strains list in **Table 1** reports all single mutants from either SL1344 or 14028 background that were used to generate P22 lysates prior to transduction to the desired SL1344 strain.

Methods, Strain construction: It should be specified that for a number of knockouts, strains from a pre-existing collection in 14028S background were used instead, citing the reference for this. It should also be specified which knockouts were used in the 14028S background.

Response: As above, we have clarified in the Results, Methods, and supplementary material which gene deletions were originally constructed in SL1344 or 14028S background.

Lines 487-488: Include a reference to the associated table of primers.

Response: We have added a reference to Table 2, which lists the primers used for strain genotyping.

Reviewer #2 (Remarks to the Author):

In this study Newson et al have developed a set of Salmonella Typhimurium multimutants that they show can be used to investigate the roles of SPI-2 T3SS effector proteins in vivo. The set of 6 multimutants were designed by grouping effectors known to interfere with specific pathogen-host functions, where possible, including intracellular replication and interference with host signal transduction. Alternatively, they were grouped due to gene location (virulence plasmid borne) or into an “unknown function” group. The approach taken to make the multimutants was thoughtful and carefully designed, for example making sure that intergenic sequences were left intact and independently making two of each multimutant to ensure that the results could be replicated. Surprisingly, 3 of the multimutants (representing almost half of the effector repertoire) replicated the WT phenotype in all the tested (murine) models. In my view this is the major weakness of the paper since the approach essentially confirmed that effectors with known functions are important but did not identify any novel effector functions.

The paper is well written and was a pleasure to read, and the experimental results were presented clearly and logically. My only writing suggestion is that the results section read more like a combined results/discussion so some of the discussion seemed rather repetitive.

Response: We appreciate the positive comments and thank the Reviewer for their time. We are pleased the Reviewer finds value in the careful design and construction of the strains, and we agree that future work might shed further light on why half of the effector repertoire appears dispensable (some commentary in the Discussion explores some possibilities). Further, the S.Tm Beta, Gamma, and Epsilon may provide starting points for further deletion of effectors with suspected redundant functions. We appreciate also the feedback regarding Discussion-relevant material in the Results section, and in the revised version of this manuscript, several of these comments have been deleted from the Results.

Minor comments

Lines 230-234: This is a little confusing and it could be misunderstood to mean that the results were similar to those seen in the spleen/liver. Maybe clearer to say that the phenotype was not apparent for most strains with only the ssaV deletion mutant and WT Delta having significantly reduced cfu levels.

Response: We are grateful for this comment, and agree the wording of this sentence could cause confusion. We have reworded this section to more clearly describe the differences between these phenotypes.

Line 265-266: The use of the word “while” seems confusing. “nevertheless” would seem more appropriate.

Response: We have reworded this sentence to be more clear.

Spelling: There is no Y in Gentamicin.

Response: We thank the reviewer for pointing out this oversight, and have corrected this spelling error.

Figures: Overall, the figures were easy to follow and well organized. With the exception of the bar graphs showing cecal pathology scores, since it was almost impossible to distinguish the four cell types in Figures 3 & 4 (Fig 5E was somewhat better but could still be improved). I would also check the use of color on graphs, which worked very well for me, but might not work for people with color blindness.

Response: We are grateful for these comments regarding accessibility and clarity of figure design, which is a priority for us as authors. To address this, we have redesigned the caecal pathology graphs to better distinguish between the four criteria (Fig 3E, Fig 4D, Fig 5E). We acknowledge also that we rely on coloured data points to distinguish between multimutant strains in a manner this is consistent throughout this manuscript and with previously published work (PMID: 40638382). These colours were selected with the aid of accessible colour pallets and were generally well received by members of our group who proof-read this manuscript. In instances where coloured data points correspond to a list of strains in the figure key (e.g. Fig 3D), we have ensured that data points and the figure key are presented in an ordered manner (i.e. the first data point within a panel section corresponds to the first strain in the figure key), which we hope assists readers with impaired colour perception. Ultimately, we acknowledge the importance of colour choices and we remain open to further feedback regarding accessible design.

Methods

Overall there seems to be good level of detail.

Mouse infections: Presumably the inocula were plated and it would be good to provide the actual cfu instead of stating 10e3 e.g. for each strain in each experiment. How many mice per group in the CI experiment – it looks like 5 but you have stated clearly elsewhere (Fig 2D). Were any of the animal experiments repeated?

Response: This also is an important point which aligns with comments made by Reviewer 1, as it is important to demonstrate that no differences exist between these strains when cultures are prepared for inoculation (i.e. mice are infected with equivalent numbers, and differences in strains recovered from mice actually correspond to differences in attenuation).

To support this, we have introduced several panels in supplementary figures to report inoculum sizes. Figure S1A (right panel) reports approximately equal values of 10^3 CFU for intraperitoneal infections described in Fig 2B. Figure S3A and S3C report approximate inocula sizes of 5×10^7 CFU for oral infections described in Figure 3 and Figure 4, respectively. The number of data points for each strain in these inoculum panels is an indicator of how many animal experiments were performed to produce these data (i.e. Fig S2A shows two data points for each multimutant strain, indicating two independent animal experiments). For the CI experiment in Fig 2D, 5 animals were indeed tested, and we have added this information to the figure legend. Ultimately, all data shown for animal experiments is a composite of multiple animal experiments combined into single graphs for clarity. The exception is Fig 2D which represents a single experiment with 5 mice, but these data phenocopy values shown in Fig S2C which are a composite of three independent animal experiments. Thus, we have a high degree of confidence in the reproducibility of these data.

Reviewer #3 (Remarks to the Author):

The authors constructed 6 multimutant strains of S. Typhimurium with multiple deletions in the cohort of SPI2 T3SS genes grouped according to the similar functions based on the previous studies. This was done to facilitate understanding of the functions of >30 of these genes encoding effector proteins. The mutants were evaluated using various mouse infection models to corroborate previous findings on these genes. Further, the mutants along with other control strains were barcode-tagged to allow simultaneous evaluation of these mutants as a pool in animal infection models.

*The study was designed carefully and performed thoroughly with the highest level of rigor, and the results were presented and summarized clearly and professionally with relevant references. Similar approaches (Chen et al 2021. doi.org/10.1016/j.chom.2021.08.012) have been employed and reported to simplify the process of assessing the functions of >30 effector genes. In that sense, the novelty of this study is limited. However, I believe the design principles based on which the genes were divided into 6 cohort groups (thus 6 multimutants) have some distinctive advantage over other similar approaches in elucidating the functions of these effector proteins as exemplified with more detailed characterization of *sopD2gtgE* double mutants.*

I believe this manuscript has a value to be published in Nature Communication. I have a few comments for consideration by the authors to improve the manuscript.

Response: We appreciate the Reviewer's comments and thank them for their time. We agree that the principal impact of the multimutant strains is the potential for screening for effector function in various experimental contexts, and we are pleased the example of characterising the *sopD2gtgE* mutant during gut infection is convincing to the Reviewer.

1. lines 157-158: the authors used two independent clones for each multimutant strain to establish if particular phenotypes are attributable to unwanted polymorphisms. I couldn't find any results shown to answer this question.

Response: This is an important point and we thank the Reviewer for raising this question. Indeed, in the original manuscript we describe the construction and validation of two independently constructed clones of each multimutant (with variability in polymorphisms arising from strain construction), yet phenotypic data *in vivo* was only shown for one clone of

each multimutant. In the revised manuscript, we expand our analysis to include the second clone of each multimutant. In Figure S1B, intraperitoneal infection of mice with the second clone of each strain shows recoverable CFU levels that phenocopy values for the first clones (Fig 2B). Similarly, we introduced fitness-neutral genetic tags to the second-clone strains and performed qRT-PCR to assess within-host fitness (as in Fig 2D), and again observed a similar degree of relative fitness compared to control strains (Fig S2C). Given the highly precise nature of qRT-PCR measurements, these data provide strong evidence that variability in polymorphisms arising from strain construction has not impacted the phenotypes of these strains, and rather that these phenotypes represent genuine attenuation of these strains due to genetic disruption of the indicated effector genes. We did not extend this comparison to oral infection, partly to limit the number of animals required for such experiments (20+ animals for reliable data), and partly because phenotypes observed in intraperitoneal (Fig 2) and orogastric (Fig 3) infections are generally similar, thus we expect both clones of each mutant to behave similarly in oral infection.

2. The authors mentioned about context-dependency (line 371) and redundancy of gene functions among the SPI2 T3SS effectors. These concepts are not something unique to these effector proteins but are a general theme in genetics. The more broad and general term for these is "genetic interactions" and redundancy would be the case for negative genetic interactions. Describing these observations in terms of genetic interactions might be a good way to understand them as a more common issues in genetics.

Response: We appreciate this comment which places effector dependency/redundancy in the broader context of bacterial genetics. We have added some further comments to the Discussion to better place our data within the context of the wider field.

3. Lines 169-173 shows the inclusion of two single deletion mutants (ssaV and efl). If the brief rationale for inclusion of these control groups is provided, it would be helpful for the readers to follow.

Response: We have expanded this comment to better describe why these control strains are included.

4. Line 388. change "reduced" to "attenuated"?

Response: We have made the suggested change.

Reviewer #4 (Remarks to the Author):

Response: We acknowledge the Reviewer's contributions and thank them for their efforts.

Point-by-point response to the reviewers' comments:

Reviewer #1:

1. *Complementation: As standard acceptable practice in bacterial research, all experimental data involving knockouts that demonstrate a phenotype should also include a parallel comparison with a complementation strain. This is essential to provide support to the reader that the phenotype is not attributed to genetic polymorphisms introduced during the genetic manipulation process or polar effects of the knockout on downstream gene expression. While the manuscript included a complementation strain for the double knockout of *gtgE* and *sopD2* from Figure 5, the majority of the manuscript focuses on the multi-mutants Alpha, Δ invG Alpha, Beta, Δ invG Beta, Gamma, Δ invG Gamma, Delta, Δ invG Delta, Epsilon, Δ invG Epsilon, Zeta and Δ invG Zeta. The data in Figures 2, 3, 4, and Figure S1 suggest the possibility of phenotypes for Alpha, Delta, Zeta, Δ invG Alpha, Δ invG Delta and Δ invG Zeta, but comparisons with the full complementation strains were not performed. Complete complementation strains for Alpha, Delta, Zeta, Δ invG Alpha, Δ invG Delta and Δ invG Zeta should be created and comparison experiments with the associated knockouts should be included. This can be accomplished using the same genomic knockin methodology used for the strains in Figure 5.*

*Response: We appreciate that complementation of deleted genes represents a common practice in bacterial genetics, and indeed represents a fundamental component of the Molecular Koch's postulates of virulence. While this manuscript does provide phenotypic data for all six multimutants and their derivatives, we chose to ultimately focus on *S. Tm* Delta to provide an example of how iterative construction allows for deductively narrowing to the individual effectors responsible for a particular phenotype (Figure 5 of this manuscript describes this process). Indeed, we describe how a double-complemented mutant that is restored for *sopD2* and *gtgE* approximates the WT strain during orogastric infection, satisfying the Molecular postulates in this instance. However, the process to chromosomally complement a multimutant strain is particularly laborious and characterised by low efficiency, and so to perform this complementation for all multimutants and their derivatives would be an enterprise well beyond the scope and requirements of this manuscript (such an exercise would take many months to achieve, and this degree of further genetic manipulation would introduce even more polymorphisms). Indeed, we maintain that the best practice for using these strains is to: 1) identify a phenotype for a particular multimutant; 2) use simpler single/double mutants to identify the responsible effector; then 3) complement this simpler mutant to restore WT-level function. We satisfy these three criteria for one strain of interest as an example of how this approach can be effective, but it remains well beyond the scope of this work to perform this analysis for all strains introduced here. We anticipate future work will take this experimental approach for particular multimutants of interest in different experimental contexts.*

Comment: *Thank you for acknowledging the importance of gene complementation. While the knockin approach is a viable strategy, if too laborious, other approaches (ex, shuttle plasmids for complementation) are available and a wide selection of reagents are accessible for this pathogen. While it is important to have a strategy for complementation of knockouts, it remains that the majority of this manuscript focuses on multimutant experiments wherein the effect of knockouts was not validated. This leads to concerns regarding interpretation of the results for the reader. It would be important to add a paragraph to the Discussion regarding other important caveats of this study. In this paragraph, the importance of complementation to rule out the possibility of polar effects on gene expression should be highlighted for the reader, and it should be explained that the majority of data in this*

manuscript would need to be confirmed by complementation analyses in order to reach conclusions.

Response: We appreciate the further comment. While we maintain our position regarding complementation and the reasonable burden of proof for linking genotype to phenotype, we can agree that polar effects or other genetic interactions may impact the phenotypes we report here. In response, we have amended the Discussion, to better equip the reader to interpret the data and findings reported here. Overall, we report several limitations or alternative explanations throughout the Discussion, which are presented alongside their corresponding arguments, rather than gathered in a distinct 'Limitations' section, per the editorial standards for this journal. We hope this is agreeable for the reviewer.

1. Multi-mutants and Analysis of SPI-1 Contribution: The experiment of multi-mutants with the additional invG knockout was missing a parallel comparison with parent multi-mutant strain within the same experiment. Matched comparisons are necessary to make conclusions about relative contribution of SPI-1 vs SPI-2 cohorts.

Response: While interesting to consider, Figure 3 already describes the phenotypes for the parent multimutants at a cost of ~45 animals, while Figure 4 describes that of the $\Delta invG$ derivatives using another ~50 animals. Further animal use to directly compare $\Delta invG$ derivatives to parent strains would not seem to be a prudent use of animal resources, and in our opinion the existing data describes these differences comprehensively and sufficiently.

Comment: *This explanation is admirable from a cost perspective, however matched comparisons within the same experiment are needed to draw conclusions from the results. This caveat should also be highlighted in the Discussion for the reader.*

Response: We would reiterate here that the scale of direct comparisons for all parent multimutants with all $\Delta invG$ derivatives is beyond the reasonable scope of the manuscript. We do provide an example in Figure 5 in which a $\Delta sopD2\Delta gtgE$ mutant is directly compared to the derivative $\Delta invG\Delta sopD2\Delta gtgE$ mutant, and these results are broadly consistent with data shown in Figures 3 and 4. Further, the Discussion is currently three pages long and includes multiple limitations that are clearly stated, so we would reaffirm that the data already describes these differences sufficiently without further direct matching and without further description of other limitations.

Figure 1D: Why is region between gogA and pipB2 missing for Beta clone 2? This should be also be mentioned in the main text, to be more clear to the reader.

Response: Due to the random nature of genomic packaging by the P22 phage, there is some variability in which region of the 14028 chromosome surrounding the deleted gene is transferred to the recipient SL1344 strain. The sequencing data for S.Tm Beta clones 1 and 2 provide a particularly striking example of this. During strain construction, gogA::cat was transferred from a 14028S donor strain to the SL1344 recipient, which resulted in different regions of the 14028S chromosome surrounding the deletion being packaged and incorporated between both recipients (i.e. clone 1 and clone 2 of the intended mutant). Our whole genome sequencing analysis uses an algorithm that can call regions of high polymorphism as deletions, which produces these large areas of reads that fail to map correctly to the SL1344 genome (visually represented as large blue bars shown for S.Tm Beta clone 2 but not for S.Tm Beta clone 1, on the left side of Fig 1D). Thus, this region is not

missing for this clone, it is simply a graphical artefact of a high degree of variation between the SL1344 and 14028S chromosomes in this genetic location.

Comment: *Thank you for the additional information. This information would be important for the reader to understand the data. This can be addressed by adding an asterisk in the figure and adding an associated note in figure caption to explain.*

Response: We appreciate the comment and agree this can cause confusion for the reader. We have added further detail to the figure legend for Fig 1D, to better explain the presence of these regions for these clones. While we could manually curate this figure to remove these regions (in the interests of reader accessibility), we feel it is more honest to leave this representation of high polymorphism intact, as it does accurately represent differences between clones of these mutants. We hope the additional detail in the figure legend, coupled with the description in the main text and provision of a full list of polymorphisms as supplementary data, will together provide the complete level of detail needed for the reader.

Figure 1D: Why is epsilon clone 2 missing the region between srfJ and spvR? This should be also be mentioned in the main text, to be more clear to the reader.

Response: As above, this is a region of high variation between the SL1344 and 14028S chromosomes that results in the algorithm tentatively flagging this region as a deletion.

Comment: *Please see response above regarding this clone.*

Response: Please see our comment above.

Reviewer #3 (Remarks to the Author):

The authors addressed all concerns I raised in the initial review sufficiently and thoroughly. I also reviewed the critiques of the other reviewers and the responses by the authors, which gave me an additional level of confidence that the revision process improved the manuscript significantly. I recommend this manuscript for publication.

Response: We are grateful to the Reviewer for their contributions to the revised manuscript, and we are pleased that all concerns have been addressed. We thank the Reviewer for their time and efforts throughout the review process.

Reviewer #4 (Remarks to the Author):

Response: We again acknowledge the Reviewer's contributions and thank them for their efforts.